# Scaling Laws for Deepfake Detection

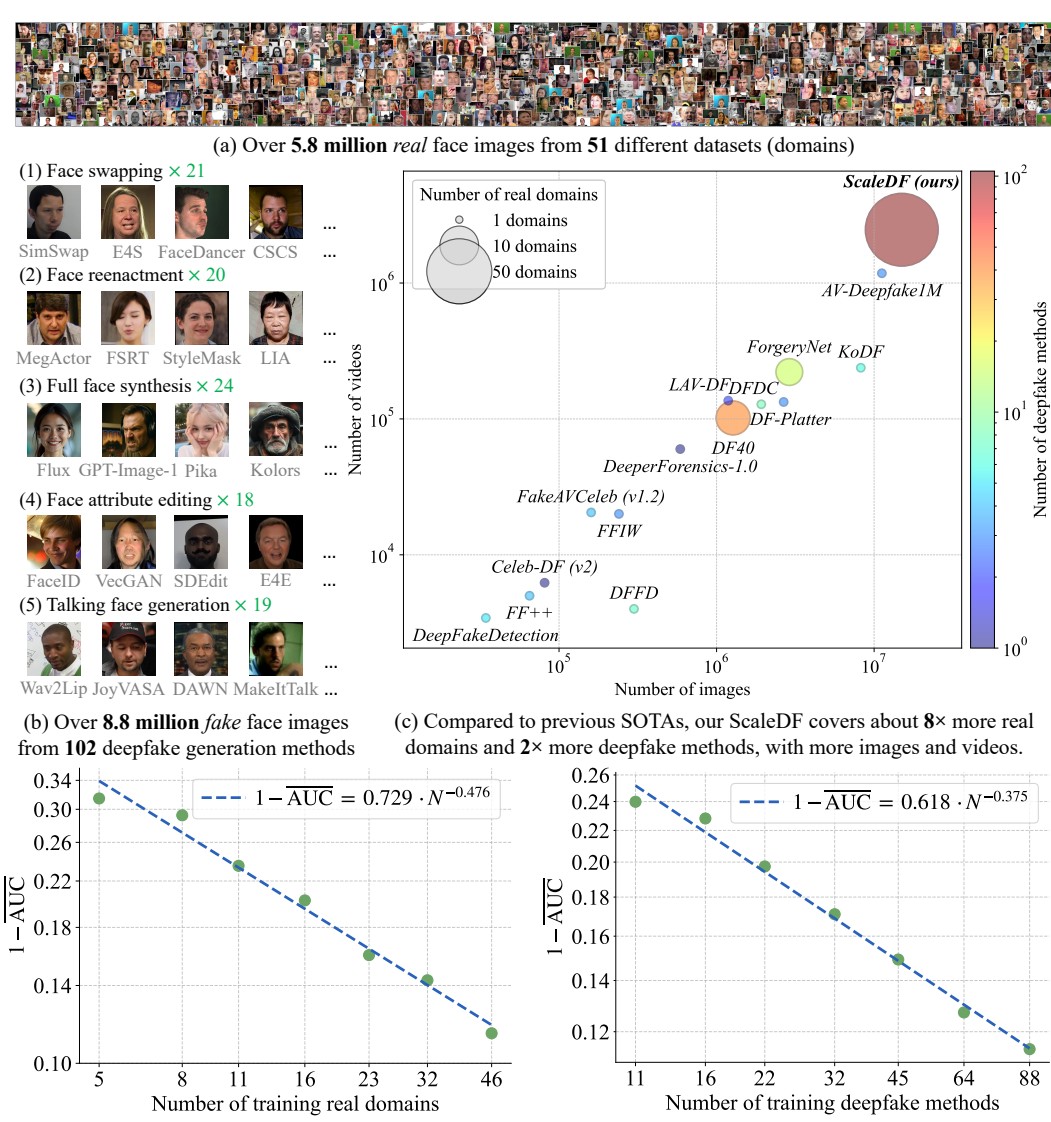

(a) Over **5.8 million** *real* face images from **51** different datasets (domains)

(b) Over **8.8 million** *fake* face images from **102** deepfake generation methods

(c) Compared to previous SOTAs, our ScaleDF covers about **8×** more real domains and **2×** more deepfake methods, with more images and videos.

(d) The observed *scaling laws* for deepfake detection

Figure 1: Illustration of ScaleDF, which is the **largest** deepfake detection dataset across four dimensions: number of real domains, number of deepfake methods, number of videos, and number of images. To ensure a fair comparison, for datasets that do not explicitly provide the number of images, we estimate image counts by sampling one frame per second from the videos. Using this dataset, we present a systematic study on scaling laws for deepfake detection and reveal several insights.

## Abstract

This paper presents a systematic study of scaling laws for the deepfake detection task. Specifically, we analyze the model performance against the number of real image domains, deepfake generation methods, and training images. Since no existing dataset meets the scale requirements for this research, we construct

ScaleDF, the largest dataset to date in this field, which contains over 5.8 million real images from 51 different datasets (domains) and more than 8.8 million fake images generated by 102 deepfake methods. Using ScaleDF, we observe power-law scaling similar to that shown in large language models (LLMs). Specifically, the average detection error follows a predictable power-law decay as either the number of real domains or the number of deepfake methods increases. This key observation not only allows us to forecast the number of additional real domains or deepfake methods required to reach a target performance, but also inspires us to counter the evolving deepfake technology in a data-centric manner. Beyond this, we examine the role of pre-training and data augmentations in deepfake detection under scaling, as well as the limitations of scaling itself.

## 1 INTRODUCTION

The rapid advancement of deepfake technology poses significant challenges to society, ranging from the dissemination of misinformation to the infringement of personal privacy, thereby necessitating the development of effective detection methods. Within the ongoing "arms race" between generation and detection, a central challenge lies in how detection models can generalize to the ever-emerging new forms of forgeries. Increasing the diversity of training data at scale has been considered a promising approach to address this issue. However, this raises a fundamental question: *is there a predictable relationship between the improvement of model performance and the growth in data scale?*

To address this question, we begin by constructing a dataset of unprecedented diversity and scale, since the scale of existing datasets is insufficient for our research. Specifically, we introduce **ScaleDF**, the largest deepfake detection dataset to date, which contains over 5.8 million real images from 51 distinct datasets (domains) and more than 8.8 million fake images generated by 102 different methods, as shown in Fig. 1 (a) and (b). Compared to previous datasets, ScaleDF includes approximately 8 times more real domains and 2 times more deepfake methods, along with a larger number of images and videos (Fig. 1 (c)). In collecting real images, we aim to incorporate all publicly available datasets featuring real faces, covering a wide range of tasks such as face detection, face recognition, and age estimation. When generating fake images, we categorize deepfake generation methods into five types: Face Swapping (FS), Face Reenactment (FR), Full Face synthesis (FF), Face attribute Editing (FE), and Talking Face generation (TF), with 21, 20, 24, 18, and 19 methods in each category. By introducing ScaleDF, the largest and most diverse deepfake detection dataset to date, we enable the discovery of predictable scaling laws, laying out the foundation for building more robust and generalizable deepfake detection systems.

In this work on scaling laws for deepfake detection, we treat the task as a binary classification problem and adopt the Vision Transformer (ViT) (Dosovitskiy et al., 2021) as the backbone. We observe that scaling laws, akin to those found in large language models (LLMs) (Kaplan et al., 2020), emerge when varying the number of training real domains or deepfake methods. Specifically, as shown in Fig. 1 (d), the detection error exhibits a power-law relationship with respect to the number of real domains or deepfake methods, *i.e.*, $1 - \overline{\text{AUC}} = A \cdot N^{-\alpha}$. More importantly, despite the large number of real domains and deepfake methods, we still see **no** signs of saturation. By varying the number of training images, we observe scaling laws similar to those found in image classification (Zhai et al., 2022). Specifically, we observe double-saturating power-law scaling with the format of $1 - \overline{\text{AUC}} = c + K \cdot (N + N_0)^{-\gamma}$. Empirically, with 46 real domains and 88 deepfake methods used in training, performance gradually saturates once the number of images exceeds $10^7$. However, we expect this saturation threshold to increase if more real domains and deepfake methods are included. With these observed scaling laws, we aim to transform deepfake-detector development from a *heuristic, trial-and-error art* into a *data-centric engineering discipline*. In addition, we investigate the impact of pre-training and data augmentation in the context of scaling. We compare the performance of the model trained on ScaleDF with that trained on relatively smaller datasets. We also discuss the limitations of scaling itself.

The key contributions of this work are:

- We introduce ScaleDF, the largest and most diverse deepfake detection dataset to date. It contains over 5.8 million real images from 51 real domains and over 8.8 million fake images generated by 102 methods, providing a foundation for scaling law study and future research in this field.

- We perform a systematic study on scaling laws for deepfake detection, discovering some predictable relationships between data scale and model performance. Specifically, we observe that the detection error follows a power-law decay as the number of real domains or deepfake methods increases.
- We conduct extensive experiments to study the effects of scaling on pre-training and data augmentation, as well as to explore its limitations. With ScaleDF, we also achieve better cross-benchmark generalization over the existing datasets.

## 2 RELATED WORKS

**Deepfake detection.** Deepfake detection aims to distinguish synthesized (or manipulated) faces from real ones. In response to the rapid advancement of face forgery techniques, many effective deepfake detection methods are proposed. For example, DiffusionFake (Chen et al., 2024) leverages diffusion models, while Hitchhikers (Foteinopoulou et al., 2024) utilizes vision-language models (VLMs) to boost detection accuracy. Additionally, several methods focus on artifacts arising from face blending (Zhou et al., 2024b; Sun et al., 2024; Zhou et al., 2024a). Frequency-based approaches also gain attention, with works exploring the use of frequency-domain information (Kashiani et al., 2025; Dutta et al., 2025; Gupta et al., 2025; Tan et al., 2024a). More recently, researchers begin exploring the adaptation and fine-tuning of pre-trained vision-language models for deepfake detection (Yan et al., 2025b; Cui et al., 2025; Yan et al., 2024). While promising, most of these methods are trained and tested on small datasets and a small number of deepfake generation techniques, which limits their usefulness in real-world scenarios with constantly emerging deepfakes. In contrast, we present a systematic scaling study in deepfake detection, aiming to investigate the underlying scaling laws. Furthermore, we are also aware of the watermark approaches such as SynthID (Kohli, 2025) and Stable Signature (Fernandez et al., 2023) for proactive detection. Our work serves as a complementary approach before these techniques evolve into universal standards.

**Scaling laws.** Scaling laws play a crucial role in guiding the training of modern foundation models (Li et al., 2025). Since their first introduction for language models (Kaplan et al., 2020; Hestness et al., 2017), numerous studies (Cherti et al., 2023; Aghajanyan et al., 2023; Tay et al., 2022; Hoffmann et al., 2022; Hu et al., 2024; Wei et al., 2022; Hernandez et al., 2021) further validate, extend, and refine scaling laws in the context of large language models (LLMs) and multimodal large language models (MLLMs). Beyond these, scaling laws are also observed in other domains, including autonomous driving (Naumann et al., 2025), image generation (Tian et al., 2024), image classification (Zhai et al., 2022), recommendation systems (Ardalani et al., 2022), and dense retrieval (Fang et al., 2024). This paper aims to extend the study of scaling laws to the domain of deepfake detection. Encouragingly, we observe power-law scaling similar to that shown in LLMs.

## 3 SCALEDF DATASET

Research on scaling laws relies heavily on large and diverse training datasets (Jain et al., 2024; Brill, 2024). However, suitable datasets for this task remain largely absent: the existing datasets either cover only a limited range of deepfake generation methods, or are too small in size. More importantly, the real domains covered in these datasets are also very limited. To solve this, this section introduces the ScaleDF dataset, designed to support research on scaling laws for deepfake detection. Specifically, we first detail the curation process of ScaleDF, and then compare it with existing datasets.

### 3.1 CURATING SCALEDF

**Real datasets collection.** Our guiding principle in collecting real face datasets is **inclusiveness**, *i.e.*, we aim to incorporate all currently publicly available datasets containing real faces. This enables us to study scaling laws with respect to the number of real datasets (domains). Specifically, we collect datasets spanning a wide range of tasks, including (1) face detection, (2) identity recognition and verification, (3) age and demographic estimation, (4) facial expression, emotion, and valence analysis, (5) audio-visual speech and speaker recognition, (6) masked-face and occlusion-robust detection, (7) multi-spectral and cross-modal biometrics, (8) talking-head synthesis, (9) spatio-temporal action localization, and (10) fairness and bias evaluation. At the same time, we intentionally exclude most of the datasets that are specifically designed for deepfake detection to prevent overfitting, as the performance of models trained on ScaleDF and evaluated on well-established deepfake detection

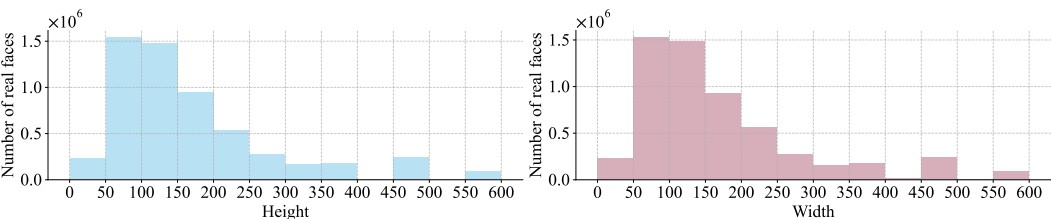

Figure 2: Statistics of heights and widths of all real faces in the ScaleDF dataset.

benchmarks remains trustworthy. The only exception is made for FaceForensics++ (Rössler et al., 2019), which is included in ScaleDF because of its wide usage in training, and we consider it too valuable a resource to exclude. Note that we do not include all images or videos from any single dataset, as we consider diversity and balance to be important. For instance, the VGGFace2 dataset (Cao et al., 2018b) contains about 3.31 million face images, of which we randomly include only 0.2 million in ScaleDF. We observe that image overlap and domain similarity between different datasets are inevitable at such a scale. Nevertheless, when splitting the training and testing datasets, we endeavor to perform so-called "cross-domain" evaluation by selecting testing datasets that are rarely covered by recent large-scale face datasets. The names, sizes, and download links of all included real datasets are provided in the Appendix (Table 6). The statistics of the heights and widths of all real faces in ScaleDF are shown in Fig. 2.

**Deepfake generation.** Inspired by a recent survey (Pei et al., 2024) and benchmark study (Yan et al., 2024), we categorize deepfake generation methods into five types and provide formal definitions:

- Face Swapping (FS): The face of one person is replaced with the face of another.
- Face Reenactment (FR): Transfer facial expressions and movements from one person to another.
- Full Face synthesis (FF): Generate entirely new faces from scratch (without visual reference).
- Face attribute Editing (FE): Modify one person's specific facial characteristics, such as age, eye, expression, hair, and nose.
- Talking Face generation (TF): Synthesize facial movements and lip synchronization from faces.

Similar to the collection of real datasets, we aim to include as many deepfake generation methods as possible to facilitate research on scaling laws along the dimension of deepfake methods. Beyond mere quantity, we also recognize the importance of diversity in two key aspects: (1) **Architectural diversity.** We cover all major architectures used in deepfake generation, including affine mapping, 3D modeling, variational autoencoders (VAE), generative adversarial networks (GANs), diffusion models, flow matching, and autoregressive (AR) models; (2) **Category balance.** Across the five categories, we collect 21, 20, 24, 18, and 19 methods respectively, ensuring a balanced representation across types. We observe that similarity between different deepfake methods is inevitable at such a scale. Nevertheless, when splitting training and testing methods, we aim to perform so-called "cross-method" evaluation by selecting testing methods (*e.g.*, GPT-Image-1 (OpenAI, 2025) and SkyReels-A1 (Qiu et al., 2025)) that are not fine-tuned or adapted from any of the training ones. During generation, we use real datasets for methods that require visual references (FS, FR, FE, TF), while for methods that do not (FF), we use either textual prompts from DiffusionDB (Wang et al., 2023b), JourneyDB (Sun et al., 2023b), and VidProM (Wang & Yang, 2024), or direct noise inputs. Note that this process uses training real datasets to generate training deepfakes and testing real datasets to generate testing deepfakes, respectively. Therefore, the ScaleDF dataset features both "cross-domain" and "cross-method" evaluation. For training and testing, we generate about 40, 000 and 2, 000 samples respectively for video-based deepfakes, and about 120, 000 and 6, 000 samples respectively for image-based deepfakes. The names, categories, architectures, and download links of all included deepfake methods are listed in the Appendix (Table 7).

**Experimental setup.** Although ScaleDF includes both videos and images, we focus on **images** for training and testing, similar to DF40 (Yan et al., 2024) and DeepfakeBench (Yan et al., 2023), since videos are essentially sequences of images (We acknowledge that temporal artifacts can be helpful for deepfake detection, but this aspect is beyond the scope of this paper.). Specifically, we uniformly sample 3 frames from each generated or real video and use them to represent the video. After that, we follow the standard face detection, alignment, and cropping procedures used in DF40 (Yan et al., 2024) and DeepfakeBench (Yan et al., 2023) to obtain the final processed faces. See Section A and B in the Appendix for more preprocessing details and visualization of processed faces. Although ScaleDF

Table 1: Compared to existing datasets, ScaleDF includes more real domains, more deepfake methods, and a larger number of videos and images, enabling scaling law research in these dimensions.

| Dataset | Real Domains | Deepfake Methods | Videos Real | Videos Fake | Images Real | Images Fake |
|---|---|---|---|---|---|---|
| DF-TIMIT (Korshunov & Marcel, 2018) | 1 | 2 | 320 | 640 | - | - |
| UADFV (Yang et al., 2019) | 1 | 1 | 49 | 49 | - | - |
| FaceForensics++ (Rossler et al., 2019) | 1 | 4 | 1,000 | 4,000 | - | - |
| DeepFakeDetection (Google, 2019) | 1 | 5 | 363 | 3,068 | - | - |
| Celeb-DF V2 (Li et al., 2020) | 1 | 1 | 590 | 5,639 | - | - |
| WildDeepfake (Zi et al., 2020) | N/A | N/A | 3,805 | 3,509 | - | - |
| DFFD (Dang et al., 2020) | 3 | 8 | 1,000 | 3,000 | 58K+ | 0.2M+ |
| DeeperForensics-1.0 (Jiang et al., 2020a) | 1 | 1 | 50K | 10K | - | - |
| DFDC (Dolhansky et al., 2020) | 1 | 8 | 23K+ | 0.1M+ | - | - |
| ForgeryNet (He et al., 2021) | 4 | 15 | 99K+ | 0.1M+ | 1.4M+ | 1.4M+ |
| FakeAVCeleb (Khalid et al., 2021) | 1 | 4 | 500 | 19.5K | - | - |
| KoDF (Kwon et al., 2021) | 1 | 6 | 62K+ | 0.1M+ | - | - |
| FFIW (Zhou et al., 2021) | 1 | 3 | 10K | 10K | - | - |
| LAV-DF (Cai et al., 2022) | 1 | 2 | 36K+ | 99K+ | - | - |
| GFW (Borji, 2022) | 2 | 3 | - | - | 30K | 15K+ |
| DF$^3$ (Ju et al., 2023) | - | 6 | - | - | - | 46K+ |
| DeepFakeFace (Song et al., 2023) | 1 | 3 | - | - | 30K | 90K |
| DF-Platter (Narayan et al., 2023) | 1 | 3 | 764 | 0.1M+ | - | - |
| DiffusionDeepfake (Chaitali et al., 2024) | - | 2 | - | - | - | 0.1M+ |
| AV-Deepfake1M (Cai et al., 2024) | 1 | 3 | 0.2M+ | 0.8M+ | - | - |
| DiFF (Cheng et al., 2024b) | 2+ | 13 | - | - | 23K+ | 0.5M+ |
| DF40 (Yan et al., 2024) | 6 | 40 | 1K+ | 0.1M+ | 0.2M+ | 1M+ |
| **ScaleDF (Ours)** | **51** | **102** | **0.9M+** | **1.4M+** | **5.8M+** | **8.8M+** |

is large and comprehensive, we remain interested in evaluating the performance of models trained on it when tested "in the wild". This is because, in real-world scenarios, many factors can influence deepfake detection performance: not only novel real domains and deepfake methods, but also image compression, face preprocessing, blending methods, and various perturbations. To this end, we select six well-established benchmarks published between 2019 and 2024, *i.e.*, DeepFakeDetection (Google, 2019), Celeb-DF V2 (Li et al., 2020), WildDeepfake (Zi et al., 2020), ForgeryNet (He et al., 2021), DeepFakeFace (Song et al., 2023), and DF40 (Yan et al., 2024), to report their direct testing performance. We adopt two commonly used evaluation metrics: the Area Under the Receiver Operating Characteristic Curve (AUC) and the Equal Error Rate (EER).

### 3.2 COMPARING SCALEDF WITH SIMILAR DATASETS

In Table 1, we compare the proposed ScaleDF dataset with existing deepfake detection datasets. The reasons they are not suitable for scaling laws research are as follows:

- **Lack of training real domains.** Most of the current deepfake datasets are sourced from only 1 to 2 real domains. We observe that only 2 large-scale datasets cover more than 3 domains; however, both have their own drawbacks: (1) ForgeryNet (He et al., 2021): although it covers 4 real domains for training, it includes only 15 outdated deepfake methods, *i.e.*, none of the latest diffusion or autoregressive models are included; and (2) DF40 (Yan et al., 2024): although it includes 6 real domains and 40 deepfake methods, only 1 real domain (FaceForensics++ (Rossler et al., 2019)) is actually used for training, and most of the fake training images come from that domain. In contrast, **ScaleDF** covers 51 real domains, 46 of which are used for training, enabling research of scaling laws across the real domain dimension.

- **Insufficient deepfake methods.** The dataset that includes the most deepfake methods so far is DF40 (Yan et al., 2024). In comparison, **ScaleDF** (1) contains twice as many deepfake generation methods, making scaling law research along the method dimension more convincing; (2) incorporates several recent and widely used methods, such as GPT-Image-1 (OpenAI, 2025) and Step1X-Edit (Liu et al., 2025), making the study of scaling laws more practical.

- **Limited number of images (videos).** Given that (1) modern deepfake datasets such as ForgeryNet (He et al., 2021) and DF40 (Yan et al., 2024) contain around $1 \sim 2$ million images, and (2) prior work in image classification (Zhai et al., 2022) has observed power-law scaling when increasing

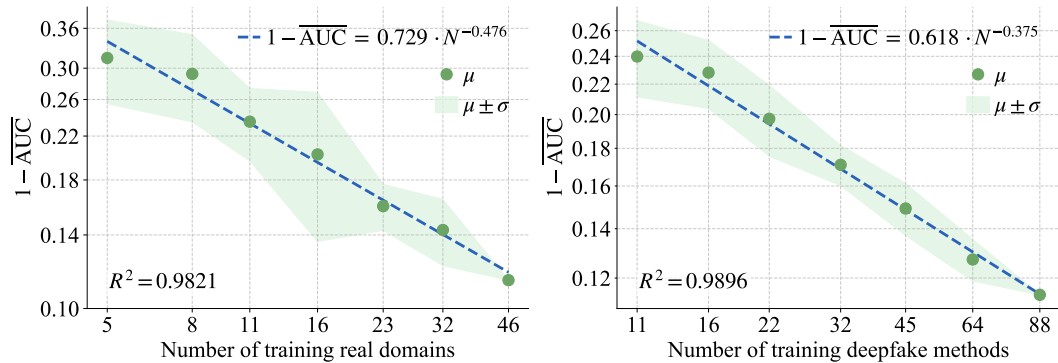

Figure 3: Left: observed power-law scaling as the number of training real domains changes; Right: observed power-law scaling as the number of training deepfake methods changes. $\mu$ represents the mean computed over 10 repetitions and 7 test datasets, while $\sigma$ denotes the variance across the 10 repetitions.

dataset size from 1 million to 10 million, we are motivated to explore whether similar scaling laws hold for deepfake detection. To this end, we scale the ScaleDF dataset to over 14 million images, *i.e.* more than $5\times$ the size of the previous largest dataset, ForgeryNet (He et al., 2021).

## 4 SCALING LAW OBSERVATIONS

In this section, we present the observed scaling laws by training models on the ScaleDF dataset and evaluating them on seven different test datasets. We report the average AUC across these datasets, denoted as $\overline{\text{AUC}}$, here, and present the corresponding scaling laws with average EER ($\overline{\text{EER}}$) in Appendix (Section C). The complete experimental results can be found in Appendix (Section E).

**Training configurations.** To eliminate the interference from more advanced deepfake detection methods in our scaling laws research, we formulate the deepfake detection task as a pure binary classification problem. We use the Vision Transformer (ViT) (Dosovitskiy et al., 2021) as the backbone, defaulting to the ViT-Base (Touvron et al., 2022) model pre-trained on ImageNet-21K (Deng et al., 2009). We perform data augmentation by first applying random image quality compression between $40\%$ and $100\%$, followed by randomly selected perturbations from AnyPattern (Wang et al., 2024b). A description of the selected perturbations and other training details are provided in Appendix (Section D and F).

**Power-law scaling is observed with respect to the number of training real domains and deepfake methods, respectively.** To study the scaling law along these dimensions, we randomly sample $N \in \{5, 8, 11, 16, 23, 32\}$ real domains from a total of 46 and $N \in \{11, 16, 22, 32, 45, 64\}$ deepfake methods from a total of 88, respectively. To reduce randomness, each sampling is repeated 10 times. For each sampled set of real domains, we train a model using all fake images and the corresponding real images from those domains; while for each sampled set of deepfake methods, we train a model using all real images and the corresponding fake images generated by those methods. Each $\mu$ in Fig. 3 represents the mean computed over 10 repetitions and 7 test datasets, while $\sigma$ denotes the variance across the 10 repetitions. Based on these empirical data points, we observe that the trend of $1 - \overline{\text{AUC}}$ with respect to the number of real domains or deepfake methods ($N$) is best described by a power law, which is the same form as originally proposed for LLMs (Kaplan et al., 2020), *i.e.*, $1 - \overline{\text{AUC}} = A \cdot N^{-\alpha}$. Using ordinary least squares (OLS), we estimate the parameters as $A = 0.729$ with $\alpha = 0.476$ for real domains, and $A = 0.618$ with $\alpha = 0.375$ for deepfake methods. Meanwhile, we calculate the coefficient of determination $R^2 = 0.9821$ and $R^2 = 0.9896$, respectively, implying that the fitted power law explains $98.21\%$ and $98.96\%$ of the variance in the observed data, thus providing an excellent description of the scaling relationship. This scaling law suggests that the performance with respect to the number of real domains and deepfake methods is **far from saturated**. To achieve higher performance, collecting more real domains and deepfake methods proves to be highly effective. Furthermore, the model's performance is, to some extent, predictable: for example, to reach an average AUC of 0.95, we may require about 300 real domains *or* 700 deepfake methods.

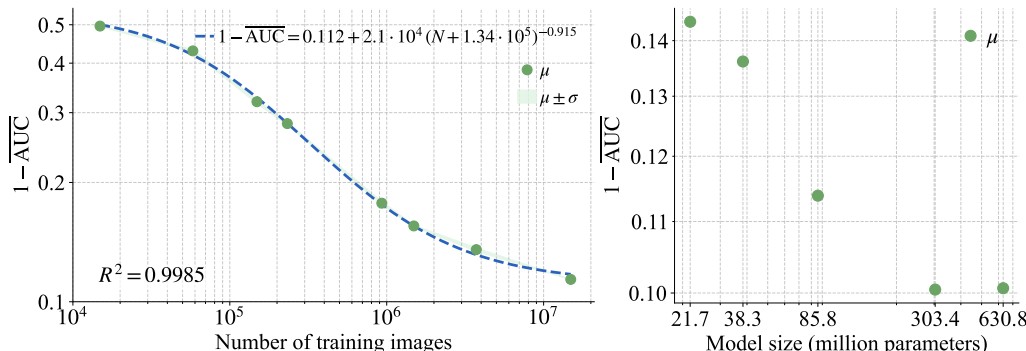

Figure 4: Left: observed double-saturating power-law scaling as the number of training images changes; $\mu$ represents the mean computed over 5 repetitions and 7 test datasets, while $\sigma$ denotes the variance across the 5 repetitions. Right: performance changes with respect to model sizes.

**The two scaling laws described above are not caused by a change in the number of real or fake images.** When conducting experiments by randomly sampling real domains, the number of real images changes (while the number of fake images remains fixed); conversely, when randomly sampling deepfake methods, the number of fake images changes (with the number of real images unchanged). A natural question is *whether the performance changes or the observed scaling laws are caused by changes in the number of real or fake images.* Our answer is **no**. To support this argument, we conduct two experiments: (1) we keep the number of fake images fixed and randomly sample $1/10$ of the real images for training; the resulting $\overline{\text{AUC}}$ decreases slightly from $0.886$ to $0.879$; (2) we keep the number of real images fixed and randomly sample $1/10$ of the fake images for training; the resulting $\overline{\text{AUC}}$ drops from $0.886$ to $0.876$ slightly. We use $1/10$ here because, when studying scaling laws, we use more than $1/10$ of the real domains or deepfake methods. These small changes in $\overline{\text{AUC}}$ suggest that the observed scaling laws are indeed related to the number of **real domains** and **deepfake methods**, rather than the number of real or fake images.

**Double-saturating power-law scaling is observed with respect to the number of training images.** To study the scaling law along this dimension, we randomly sample subsets of the ScaleDF training set at the following proportions: $1/4, 1/10, 1/16, 1/64, 1/100, 1/256, 1/1000$. Note that this differs from the previous section, *i.e.*, here we sample real and fake images simultaneously. To reduce randomness, each sampling is repeated 5 times. Each $\mu$ in Fig. 4 (left) represents the mean computed over 5 repetitions and 7 test datasets, while $\sigma$ denotes the variance across the 5 repetitions. Based on these empirical data points, we observe that the trend of $1 - \overline{\text{AUC}}$ with respect to the number of training images ($N$) is best described by a double-saturating power law, which is the same form as originally proposed in image classification (Zhai et al., 2022), *i.e.*, $1 - \overline{\text{AUC}} = c + K \cdot (N + N_0)^{-\gamma}$. Using ordinary least squares (OLS), we estimate the parameters as $c = 0.112$, $K = 2.1 \cdot 10^4$, $N_0 = 1.34 \cdot 10^5$, and $\gamma = 0.915$. Meanwhile, we calculate the coefficient of determination $R^2 = 0.9985$, implying that the fitted power law explains $99.85\%$ of the variance in the observed data, thus providing an excellent description of the scaling relationship. This scaling law suggests that, with 46 real domains and 88 deepfake methods, performance gradually saturates once the total number of training images exceeds $10^7$. To further improve performance, blindly collecting more images from the same real domains or generating more from the same deepfake methods may **not be effective**. However, this does **not** imply that datasets larger than $10^7$ images offer no further benefit for deepfake detection. Rather, as we scale up the number of real domains and deepfake methods, more training data will still be essential.

**When training on ScaleDF, model size scaling appears to saturate at around 300 million parameters.** Beyond the aforementioned scaling laws, we are also interested in scaling along the model size dimension, *i.e.*, how large a model ScaleDF can support. Specifically, we select five different model sizes, denoted as ViT-S (21.7M), ViT-M (38.3M), ViT-B (85.8M), ViT-L (303.4M), and ViT-H (630.8M). Each $\mu$ in Fig. 4 (right) represents the mean computed over 7 test datasets. Performance consistently improves from ViT-S (21.7M) to ViT-L (303.4M), but saturates thereafter, as further scaling to ViT-H (630.8M) yields no gains. This observation does **not** suggest that models with more than 300M parameters bring no additional gain for the deepfake detection task; rather, it indicates the maximum model size currently supported by the ScaleDF dataset. In the future, as we scale up real domains and deepfake methods, larger models may yield better performance.

Table 2: Comparison of using different pre-training models: similar performance observed.

| AUC | DFD | CDF V2 | Wild | Forgery. | DFF | DF40 | ScaleDF | Mean |
|---|---|---|---|---|---|---|---|---|
| ImageNet | 0.793 | 0.915 | 0.815 | 0.824 | 0.909 | 0.980 | 0.968 | 0.886 |
| CLIP | 0.795 | 0.894 | 0.802 | 0.848 | 0.954 | 0.979 | 0.981 | 0.893 |
| SigLIP 2 | 0.750 | 0.890 | 0.785 | 0.855 | 0.951 | 0.986 | 0.983 | 0.886 |

Table 3: Effectiveness of image quality compression (QC) and perturbations (PT).

| AUC | DFD | CDF V2 | Wild | Forgery. | DFF | DF40 | ScaleDF | Mean |
|---|---|---|---|---|---|---|---|---|
| N/A | 0.667 | 0.805 | 0.781 | 0.756 | 0.808 | 0.937 | 0.975 | 0.818 |
| QC | 0.774 | 0.914 | 0.802 | 0.770 | 0.932 | 0.978 | 0.976 | 0.878 |
| QC + PT | 0.793 | 0.915 | 0.815 | 0.824 | 0.909 | 0.980 | 0.968 | 0.886 |

## 5 ADDITIONAL OBSERVATIONS WITH SCALING

Beyond scaling laws, we identify additional insights as deepfake detection is scaled up. We report AUC results here, with EER presented in Appendix (Section G). Specifically, we observe that:

**Different pre-trained models exhibit similar performance with the ScaleDF.** Several works (Ojha et al., 2023; Yan et al., 2024; 2025a) have reported that, on **small-scale** datasets, fine-tuning pre-trained vision-language models yields better performance than fine-tuning pre-trained image classification models. Here, we aim to examine whether this claim holds when training on the **large-scale** ScaleDF dataset, which contains over 14 million images. Our findings indicate that the answer is **no**. Specifically, we compare three types of pre-trained ViT-Base models: (1) image classification on ImageNet-21K, (2) CLIP (Radford et al., 2021), and (3) SigLIP 2 (Tschannen et al., 2025). Experiments in Table 2 show that there are no significant performance differences (less than 1% in AUC) across different pre-training methods. Nevertheless, we also observe that without pre-training, convergence is very slow, and thus pre-training remains necessary for the ScaleDF.

**Data augmentation remains important in the context of scaling.** In this section, we investigate the importance of data augmentation in training on the large-scale ScaleDF dataset. A common understanding is that data augmentation serves as a remedy for data scarcity, especially in scenarios where models are data-hungry. For example, DeiT (Touvron et al., 2021) demonstrates that, with appropriate data augmentation, it is possible to achieve competitive performance using only ImageNet-1k, compared to models pre-trained on hundreds of millions of images. However, we observe that for the deepfake detection task, data augmentation remains important even when sufficient training data is available. From Table 3, we observe that: (1) random image quality compression can significantly improve performance, for example, increasing the AUC on Celeb-DF V2 (Li et al., 2020) from 0.805 to 0.914; and (2) random perturbations enhance robustness on test sets with strong perturbations, such as ForgeryNet (He et al., 2021). These improvements also suggest that deepfake detection relies, to some extent, on low-level features.

**Scaling proves essential for achieving satisfactory cross-benchmark performance.** Unlike previous dataset papers, such as ForgeryNet (He et al., 2021), DF40 (Yan et al., 2024), and DiFF (Cheng et al., 2024b), which mainly focus on proposing new comprehensive benchmarks and conducting extensive evaluations, we emphasize cross-benchmark performance, *i.e.*, training models on ScaleDF and testing them on other well-established benchmarks. This better reflects whether researchers and engineers can train a model on the proposed dataset and directly deploy it in real-world production environments. In Table 4, we compare the cross-benchmark capability of the proposed ScaleDF with that of existing datasets. It is observed that: **(1)** Although recent datasets such as DF40 (Yan et al., 2024) are large-scale and cover many deepfake methods, they still do not perform well in cross-benchmark testing. We infer that this is due to insufficient coverage of real domains. **(2)** ForgeryNet (He et al., 2021) does not achieve consistently good performance across all datasets, likely because it lacks coverage of some recent deepfake generation methods. **(3)** As expected, small-scale datasets do not perform well on cross-benchmark settings, as they cover neither a wide range of real domains nor diverse deepfake methods.

**Scaling is important but not all you need, and methodological innovations are still necessary.** As an ablation study, we remove FaceForensics++ (Rössler et al., 2019) from ScaleDF, *i.e.*, reducing one real domain and four deepfake methods, and train a new model on the modified dataset. From Table 5, we observe a performance drop on older benchmarks, while the performance on newer

Table 4: Comparison of cross-benchmark performance: with scaling, we achieve the best performance.

| | AUC | DFD | CDF V2 | Wild | Forgery. | DFF | DF40 | ScaleDF |
|---|---|---|---|---|---|---|---|---|
| Training sets | DFD | − | 0.802 | 0.738 | 0.610 | 0.545 | 0.594 | 0.524 |
| | CDF V2 | 0.709 | − | 0.777 | 0.640 | 0.599 | 0.665 | 0.612 |
| | Wild | 0.643 | 0.757 | − | 0.555 | 0.489 | 0.562 | 0.556 |
| | Forgery. | 0.813 | 0.913 | 0.821 | − | 0.687 | 0.833 | 0.657 |
| | DFF | 0.583 | 0.568 | 0.552 | 0.578 | − | 0.669 | 0.622 |
| | DF40 | 0.587 | 0.800 | 0.684 | 0.677 | 0.607 | − | 0.667 |
| | ScaleDF | 0.793 | 0.915 | 0.815 | 0.824 | 0.909 | 0.980 | − |

Table 5: Comparison of whether includes FaceForensics++ (Rössler et al., 2019) in the ScaleDF.

| AUC | DFD | CDF V2 | Wild | Forgery. | DFF | DF40 | ScaleDF | Mean |
|---|---|---|---|---|---|---|---|---|
| w/o FF++ | 0.758 | 0.868 | 0.796 | 0.807 | 0.905 | 0.978 | 0.969 | 0.869 |
| w FF++ | 0.793 | 0.915 | 0.815 | 0.824 | 0.909 | 0.980 | 0.968 | 0.886 |

benchmarks remains unchanged. This implies that simply adding more new deepfake methods does not significantly improve performance on fundamentally different, older ones. To enhance performance on such benchmarks, it is necessary to include deepfake methods similar to those used in older datasets (*e.g.*, those found in FaceForensics++). This further implies that, even though ScaleDF includes a wide range of deepfake methods, models trained with simple binary classification may not generalize well to totally different ones that could appear in the future. This suggests the need for algorithms that (1) can learn the underlying essence of forgery from a large number of deepfake methods, and (2) generalize far beyond what simple binary classification allows. We hope that the ScaleDF dataset provides sufficient resources to explore and develop such algorithms, and we call for efforts from the community.

## 6  CONCLUSION

In this paper, we introduce **ScaleDF**, the most comprehensive deepfake detection dataset to date, spanning 51 real domains, 102 deepfake methods, and more than 14 million face images. Leveraging this unprecedented scale, we conduct a systematic study of **scaling laws** in deepfake detection. Our primary finding is that detection performance follows predictable power-law scaling relationships. Specifically, we show that detection error decreases as a power-law function of the number of training real domains or deepfake methods, with no signs of saturation. This discovery shifts the development of deepfake detectors from a heuristic-driven process to a more predictable, data-centric engineering discipline. We further identify a double-saturating power law with respect to the number of training images, suggesting that when the diversity of sources is fixed, the benefit of simply increasing data quantity eventually diminishes. Strategically, we should focus on the diversity and comprehensiveness of the evolving deepfake methods when building such a database. In addition, our large-scale experiments yield several practical insights: data augmentation remains crucial for robustness, even with massive datasets, and models trained on ScaleDF achieve better cross-benchmark generalization. However, our work also highlights that scaling alone is not a panacea; generalizing to fundamentally novel or unseen forgery types remains a challenge, underscoring the need for innovation in detection algorithms. By building ScaleDF, we aim to provide a resource for the research community to explore these frontiers and encourage future efforts to develop algorithms capable of learning more fundamental forgery representations from a large and diverse dataset.

**Disclaimer for scaling law research.** We emphasize that the scaling laws presented in this work are empirical observations specific to our experimental setup. In the study of scaling laws, fitted parameters and empirical findings are known to vary depending on choices of hyperparameter configurations, optimization methods, model architectures, data quality, and other experimental conditions (Cherti et al., 2023; Bahri et al., 2024). The specific exponents and constants we report are contingent on our use of the ViT architecture, the composition of the ScaleDF dataset, and our chosen training and data augmentation protocols. As such, these laws should be interpreted as descriptive models for deepfake detection scaling within this context, rather than as universal, fundamental constants. The primary value of our findings is the demonstration that predictable scaling is achievable in this domain, providing a data-centric framework for future development. Future work exploring different architectures or data compositions would likely yield different scaling coefficients.

## ETHICS STATEMENT

This work aims to advance deepfake detection and foster Generative AI safety. We adhere to the license terms of all data and models in constructing ScaleDF. There are a few open issues and important considerations around fairness and bias. Please refer to Appendix (Section H) for a detailed discussion.

## REPRODUCIBILITY STATEMENT

A detailed description of the dataset curation process is provided in Section 3, along with complete lists and links to all source real datasets (Table 6) and deepfake generation methods (Table 7). The data preprocessing pipeline, which includes face detection, alignment, and cropping, is described in Appendix (Section A). All experimental settings, such as model configurations, data augmentation strategies, training hyperparameters, and computational infrastructure, are documented in Section 4 (Training Configurations) and Appendix (Section D and F). To support verification of our scaling law findings, we provide complete numerical results for all experiments, including Equal Error Rate (EER) metrics, in Appendix (Section C and E).

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

Table 6: Real datasets included in the ScaleDF, with the testing ones highlighted in �.

| No. | Dataset | Format | Vol. | Link | No. | Dataset | Format | Vol. | Link |
|---|---|---|---|---|---|---|---|---|---|
| 0 | GRID (Cooke et al., 2006) | Video | 16K+ | Link | 26 | AVSpeech (Ephrat et al., 2018) | Video | 0.1M+ | Link |
| 1 | MORPH-2 (Ricanek & Tesafaye, 2006) | Image | 49K+ | Link | 27 | ExpW (Zhang et al., 2018) | Image | 67K+ | Link |
| 2 | LFW (Huang et al., 2008) | Image | 13K+ | Link | 28 | IMDb-Face (Wang et al., 2018) | Image | 0.2M+ | Link |
| 3 | Multi-PIE (Gross et al., 2008) | Image | 0.1M+ | Link | 29 | RAVDESS (Livingstone & Russo, 2018) | Video | 2.4K+ | Link |
| 4 | GENKI-4K (MPLab, 2009) | Image | 3.8K+ | Link | 30 | Tufts Face (Panetta et al., 2018) | Image | 2.9K+ | Link |
| 5 | YouTubeFaces (Wolf et al., 2011) | Image | 0.2M+ | Link | 31 | VGGFace2 (Cao et al., 2018b) | Image | 0.2M+ | Link |
| 6 | IMFDB (Setty et al., 2013) | Image | 10K+ | Link | 32 | Celeb-500K (Cao et al., 2018a) | Image | 0.2M+ | Link |
| 7 | Adience (Eidinger et al., 2014) | Image | 18K+ | Link | 33 | IJB-C (Maze et al., 2018) | Image | 0.2M+ | Link |
| 8 | CACD (Chen et al., 2014) | Image | 0.1M+ | Link | 34 | VoxCeleb2 (Chung et al., 2018) | Video | 0.5M+ | Link |
| 9 | CASIA-WebFace (Yi et al., 2014) | Image | 0.2M+ | Link | 35 | Aff-Wild2 (Kollias & Zafeiriou, 2019) | Image | 0.1M+ | Link |
| 10 | CREMA-D (Cao et al., 2014) | Image | 19K+ | Link | 36 | FFHQ (Karras et al., 2019) | Image | 51K+ | Link |
| 11 | FaceScrub (Ng & Winkler, 2014) | Image | 41K+ | Link | 37 | FaceForensics++ (Rössler et al., 2019) | Video | 1K | Link |
| 12 | 300VW (Shen et al., 2015) | Video | 0.9K+ | Link | 38 | BUPT-CBFace (Zhang & Deng, 2020) | Image | 0.2M+ | Link |
| 13 | CelebA (Liu et al., 2015) | Image | 0.1M+ | Link | 39 | DFEW (Jiang et al., 2020b) | Video | 8.1K+ | Link |
| 14 | AFAD (Niu et al., 2016) | Image | 0.1M+ | Link | 40 | MEAD (Wang et al., 2020) | Image | 0.2M+ | Link |
| 15 | CFPW (Sengupta et al., 2016) | Image | 5.5K+ | Link | 41 | MMA (MahmoudiMA, 2020) | Image | 79K+ | Link |
| 16 | WIDER FACE (Yang et al., 2016) | Image | 20K+ | Link | 42 | SAMM V3 (Yap et al., 2020) | Image | 11K+ | Link |
| 17 | AffectNet (Mollahosseini et al., 2017) | Image | 30K+ | Link | 43 | FairFace (Karkkainen & Joo, 2021) | Image | 79K+ | Link |
| 18 | AgeDB (Moschoglou et al., 2017) | Image | 15K+ | Link | 44 | Glint360K (An et al., 2021) | Image | 0.2M+ | Link |
| 19 | MAFA (Ge et al., 2017) | Image | 0.5K+ | Link | 45 | SpeakingFaces (Abdrakhmanova et al., 2021) | Image | 0.2M+ | Link |
| 20 | RAF-DB (Li et al., 2017) | Image | 9.8K+ | Link | 46 | Wiki-Faces (Ford & Shao, 2021) | Image | 39K+ | Link |
| 21 | UMDFaces (Bansal et al., 2017) | Image | 0.2M+ | Link | 47 | Asian-Celeb (kyquac, 2022) | Image | 0.2M+ | Link |
| 22 | UTKFace (Zhang et al., 2017b) | Image | 23K+ | Link | 48 | CelebV-HQ (Zhu et al., 2022) | Video | 17K+ | Link |
| 23 | AFEW-VA (Kossaifi et al., 2017) | Image | 22K+ | Link | 49 | RMFD (Wang et al., 2023a) | Image | 22K+ | Link |
| 24 | MegaAge (Zhang et al., 2017a) | Image | 40K+ | Link | 50 | FaceVid-1K (Di et al., 2024) | Video | 0.7K+ | Link |
| 25 | AVA (Gu et al., 2018) | Video | 0.1M+ | Link | | | | | |

## A  DATA PREPROCESSING PIPELINE

In this section, we describe the data preprocessing pipeline, which we follow from DF40 (Yan et al., 2024) and DeepfakeBench (Yan et al., 2023), consisting of face detection, alignment, and cropping.

**Face detection.** For each input image, the preprocessing begins with locating the facial region. We utilize the frontal face detector from the Dlib library (King, 2009), a widely adopted method based on Histogram of Oriented Gradients (HOG) features. The detector identifies all potential facial bounding boxes in the image. To focus on the primary subject, when multiple faces are detected, we select the one with the largest bounding box area. If no face is detected, the image is discarded.

**Alignment.** Following face detection, a facial alignment procedure is applied to standardize the pose and scale of the detected faces. This process involves two key steps: (1) Landmark localization. We employ Dlib's pre-trained 81-point facial landmark predictor to accurately identify key facial features. From the detected landmarks, we select five critical points used for alignment: the centers of the left and right eyes, the tip of the nose, and the left and right corners of the mouth. (2) Transformation estimation. A similarity transformation is computed to map the selected landmarks to a predefined set of canonical coordinates that represent an ideal, upright facial configuration.

**Cropping.** The final step leverages the alignment information to crop the face from the original image. The computed affine transformation matrix is applied to the full image, simultaneously rotating, scaling, and translating it to center the aligned face within a new canvas. A scaling parameter, set to 1.3 in our implementation, defines the cropping boundary to avoid overly tight crops and preserve contextual facial regions such as the forehead, chin, and hair. The resulting aligned and cropped image is then resized to a fixed resolution of $256 \times 256$ pixels for subsequent processing.

## B  VISUALIZATION OF PROCESSED FACES

As illustrated in Tables 20 to 35, five processed faces are randomly sampled for each real dataset (domain) and deepfake method to provide visual examples.

Table 7: Deepfake methods included in the ScaleDF, with the testing ones highlighted in ▨.

| No. | Method | Cat. | Arch. | Link | No. | Method | Cat. | Arch. | Link |
|---|---|---|---|---|---|---|---|---|---|
| 0 | Faceswap (Earl, 2015) | FS | Affine | Code | 51 | FLUX.1 [dev] (Black, 2024) | FF | Diff. | Code |
| 1 | FaceSwap (Kowalski, 2016) | FS | 3D | Code | 52 | CogView4 (Zheng et al., 2024) | FF | Diff. | Code |
| 2 | DeepFakes (deepfakes, 2017) | FS | VAE | Code | 53 | CogView3 (Zheng et al., 2024) | FF | Diff. | Code |
| 3 | FSGAN (Nirkin et al., 2019) | FS | GAN | Code | 54 | Kolors (Kolors, 2024) | FF | Diff. | Code |
| 4 | SimSwap (Chen et al., 2020) | FS | GAN | Code | 55 | Hunyuan-DiT (Li et al., 2024c) | FF | Diff. | Code |
| 5 | HifiFace (Wang et al., 2021c) | FS | 3D | Code | 56 | LTX-Video (HaCohen et al., 2024) | FF | Diff. | Code |
| 6 | InfoSwap (Gao et al., 2021) | FS | GAN | Code | 57 | HunyuanVideo (Kong et al., 2024) | FF | Diff. | Code |
| 7 | UniFace (Xu et al., 2022a) | FS | GAN | Code | 58 | Pika (Pika, 2024) | FF | N/A | Source |
| 8 | MobileFaceSwap (Xu et al., 2022b) | FS | GAN | Code | 59 | GPT-Image-1 (OpenAI, 2025) | FF | N/A | Source |
| 9 | E4S (Liu et al., 2022) | FS | GAN | Code | 60 | Janus-Pro (Chen et al., 2025a) | FF | AR | Code |
| 10 | GHOST (Groshev et al., 2022) | FS | GAN | Code | 61 | SimpleAR (Wang et al., 2025) | FF | AR | Code |
| 11 | BlendFace (Shiohara et al., 2023) | FS | GAN | Code | 62 | Wan-T2V (Wan-Team, 2025) | FF | Diff. | Code |
| 12 | FaceDancer (Rosberg et al., 2023) | FS | GAN | Code | 63 | Pyramid Flow (Jin et al., 2025) | FF | AR | Code |
| 13 | 3DSwap (Li et al., 2023) | FS | 3D | Code | 64 | CogVideoX (Yang et al., 2025b) | FF | Diff. | Code |
| 14 | Inswapper (Wang, 2023) | FS | GAN | Code | 65 | SDEdit (Meng et al., 2021) | FE | Diff. | Code |
| 15 | FaceAdapter (Han et al., 2024) | FS | Diff. | Code | 66 | E4E (Tov et al., 2021) | FE | GAN | Code |
| 16 | CSCS (Huang et al., 2024) | FS | GAN | Code | 67 | EDICT (Wallace et al., 2022) | FE | Diff. | Code |
| 17 | REFace (Baliah et al., 2024) | FS | Diff. | Code | 68 | DiffusionCLIP (Kim et al., 2022) | FE | Diff. | Code |
| 18 | FaceFusion (Wang, 2024) | FS | GAN | Code | 69 | VecGAN (Dalva et al., 2022) | FE | GAN | Code |
| 19 | InstantID (Wang et al., 2024a) | FS | Diff. | Code | 70 | InstructPix2Pix (Brooks et al., 2023) | FE | Diff. | Code |
| 20 | DiffFace (Kim et al., 2025) | FS | Diff. | Code | 71 | IP-Adapter (Ye et al., 2023) | FE | Diff. | Code |
| 21 | Face2Face (Thies et al., 2016) | FR | 3D | Code | 72 | MaskFaceGAN (Pernuš et al., 2023) | FE | GAN | Code |
| 22 | FOMM (Siarohin et al., 2019) | FR | Affine | Code | 73 | SDFlow (Li et al., 2024a) | FE | GAN | Code |
| 23 | NeuralTextures (Thies et al., 2019) | FR | 3D | Code | 74 | EmoStyle (Azari & Lim, 2024) | FE | GAN | Code |
| 24 | OneShot (Wang et al., 2021b) | FR | 3D | Code | 75 | Triplane (Bilecen et al., 2025) | FE | GAN | Code |
| 25 | Face-Vid2Vid (Zheng, 2021) | FR | 3D | Code | 76 | FaceID (Kwai, 2024) | FE | Diff. | Code |
| 26 | TPSMM (Zhao et al., 2022) | FR | Affine | Code | 77 | AnySD (Yu et al., 2024) | FE | Diff. | Code |
| 27 | DaGAN (Hong et al., 2022) | FR | GAN | Code | 78 | MagicFace (Wei et al., 2025a) | FE | Diff. | Code |
| 28 | LIA (Wang et al., 2022) | FR | Affine | Code | 79 | RigFace (Wei et al., 2025b) | FE | Diff. | Code |
| 29 | AMatrix (Bounareli et al., 2022) | FR | GAN | Code | 80 | FluxEdit (Paul, 2025) | FE | Diff. | Code |
| 30 | StyleMask (Bounareli et al., 2023a) | FR | GAN | Code | 81 | RFInversion (Rout et al., 2025) | FE | Diff. | Code |
| 31 | MRFA (Tao et al., 2023) | FR | Affine | Code | 82 | Step1X-Edit (Liu et al., 2025) | FE | LLM | Code |
| 32 | HyperReenact (Bounareli et al., 2023b) | FR | GAN | Code | 83 | MakeItTalk (Zhou et al., 2020) | TF | GAN | Code |
| 33 | MCNet (Hong & Xu, 2023) | FR | Affine | Code | 84 | Wav2Lip (Prajwal et al., 2020) | TF | GAN | Code |
| 34 | CVTHead (Ma et al., 2024a) | FR | 3D | Code | 85 | Audio2Head (Wang et al., 2021a) | TF | GAN | Code |
| 35 | FollowYourEmoji (Ma et al., 2024b) | FR | Diff. | Code | 86 | SadTalker (Zhang et al., 2022) | TF | 3D | Code |
| 36 | LivePortrait (Guo et al., 2024) | FR | 3D | Code | 87 | Video-Retalking (Cheng et al., 2022) | TF | GAN | Code |
| 37 | Megactor (Yang et al., 2024) | FR | Diff. | Code | 88 | DreamTalk (Ma et al., 2023) | TF | Diff. | Code |
| 38 | G3FA (Javanmardi et al., 2024) | FR | 3D | Code | 89 | IP_LAP (Zhong et al., 2023) | TF | GAN | Code |
| 39 | FSRT (Rochow et al., 2024) | FR | Affine | Code | 90 | Real3DPortrait (Ye et al., 2024) | TF | 3D | Code |
| 40 | SkyReels-A1 (Qiu et al., 2025) | FR | Diff. | Code | 91 | FLOAT (Ki et al., 2024) | TF | Flow | Code |
| 41 | StyleGAN2 (Karras et al., 2020) | FF | GAN | Code | 91 | JoyVASA (Cao et al., 2024) | TF | Diff. | Code |
| 42 | VQGAN (Esser et al., 2021) | FF | GAN | Code | 93 | DAWN (Cheng et al., 2024a) | TF | Diff. | Code |
| 43 | StyleGAN3 (Karras et al., 2021) | FF | GAN | Code | 94 | AniTalker (Liu et al., 2024) | TF | Diff. | Code |
| 44 | StyleGAN-XL (Sauer et al., 2022) | FF | GAN | Code | 95 | AniPortrait (Wei et al., 2024) | TF | 3D | Code |
| 45 | SD2.1 (Rombach et al., 2022) | FF | Diff. | Code | 96 | EDTalk (Tan et al., 2024b) | TF | 3D | Code |
| 46 | SD1.5 (Rombach et al., 2022) | FF | Diff. | Code | 97 | Diff2Lip (Mukhopadhyay et al., 2024) | TF | Diff. | Code |
| 47 | SDXL (Podell et al., 2023) | FF | Diff. | Code | 98 | JoyHallo (Shi et al., 2024) | TF | Diff. | Code |
| 48 | PixArt-Alpha (PixArt, 2023) | FF | Diff. | Code | 99 | Ditto (Li et al., 2024b) | TF | Diff. | Code |
| 49 | Midjourney (Sun et al., 2023a) | FF | N/A | Source | 100 | KDTalker (Yang et al., 2025a) | TF | Diff. | Code |
| 50 | SD3.5 (Esser et al., 2024) | FF | Diff. | Code | 101 | Echomimic (Chen et al., 2025b) | TF | Diff. | Code |

## C    Scaling laws with $\overline{\text{EER}}$

In this section, we investigate whether similar scaling laws hold under a different evaluation metric, *i.e.*, $\overline{\text{EER}}$. To this end, we replicate the scaling law analysis presented in the main paper. As shown in Fig. 5, **similar scaling laws** can also be observed in terms of $\overline{\text{EER}}$ (The complete experimental results can be found in Appendix (Section E)). Specifically, we conclude that:

**Power-law scaling is observed with respect to the number of training real domains and deepfake methods, respectively.** To study the scaling law along these dimensions, we randomly sample

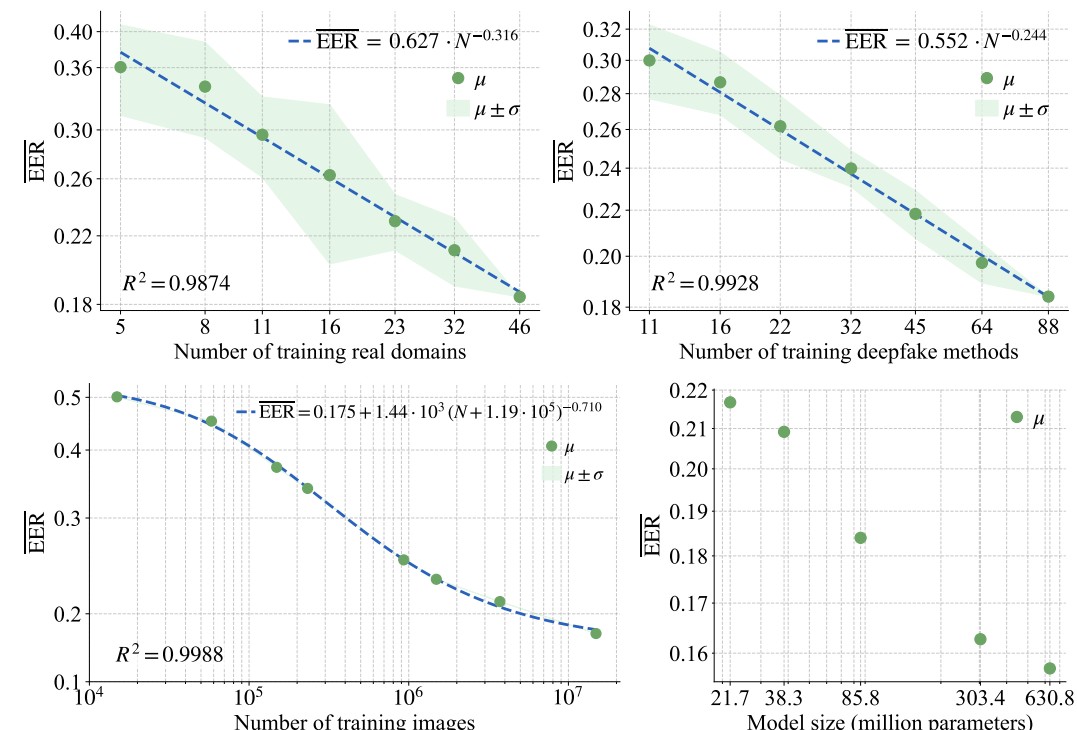

Figure 5: Left Top: observed power-law scaling as the number of training real domains changes; Right Top: observed power-law scaling as the number of training deepfake methods changes; Left Bottom: observed double-saturating power-law scaling as the number of training images changes; Right Bottom: performance changes with respect to model sizes. $\mu$ represents the mean computed over repetitions and test datasets, while $\sigma$ denotes the variance across repetitions.

$N \in \{5, 8, 11, 16, 23, 32\}$ real domains from a total of $46$ and $N \in \{11, 16, 22, 32, 45, 64\}$ deepfake methods from a total of $88$, respectively. To reduce randomness, each sampling is repeated $10$ times. For each sampled set of real domains, we train a model using all fake images and the corresponding real images from those domains; while for each sampled set of deepfake methods, we train a model using all real images and the corresponding fake images generated by those methods. Each $\mu$ in Fig. 5 (Top) represents the mean computed over $10$ repetitions and $7$ test datasets, while $\sigma$ denotes the variance across the $10$ repetitions. Based on these empirical data points, we observe that the trend of $\overline{\text{EER}}$ with respect to the number of real domains or deepfake methods ($N$) is best described by a power law, which is the same form as originally proposed for LLMs (Kaplan et al., 2020), *i.e.*, $\overline{\text{EER}} = A \cdot N^{-\alpha}$. Using ordinary least squares (OLS), we estimate the parameters as $A = 0.627$ with $\alpha = 0.316$ for real domains, and $A = 0.552$ with $\alpha = 0.244$ for deepfake methods. Meanwhile, we calculate the coefficient of determination $R^2 = 0.9874$ and $R^2 = 0.9928$, respectively, implying that the fitted power law explains $98.74\%$ and $99.28\%$ of the variance in the observed data, thus providing an excellent description of the scaling relationship. This scaling law suggests that the performance with respect to the number of real domains and deepfake methods is **far from saturated**. To achieve higher performance, collecting more real domains and deepfake methods proves to be highly effective. Furthermore, the model's performance is, to some extent, predictable: for example, to reach an average EER of $0.10$, we may require about $340$ real domains or $1100$ deepfake methods.

**The two scaling laws described above are not caused by a change in the number of real or fake images.** When conducting experiments by randomly sampling real domains, the number of real images changes (while the number of fake images remains fixed); conversely, when randomly sampling deepfake methods, the number of fake images changes (with the number of real images unchanged). A natural question is *whether the performance changes or the observed scaling laws are caused by changes in the number of real or fake images.* Our answer is **no**. To support this argument, we conduct two experiments: (1) we keep the number of fake images fixed and randomly sample $1/10$ of the real images for training; the resulting $\overline{\text{EER}}$ decreases slightly from $0.184$ to $0.193$; (2) we keep the number of real images fixed and randomly sample $1/10$ of the fake images for training;

the resulting $\overline{\text{EER}}$ drops from $0.184$ to $0.198$ slightly. We use $1/10$ here because, when studying scaling laws, we use more than $1/10$ of the real domains or deepfake methods. These small changes in $\overline{\text{EER}}$ suggest that the observed scaling laws are indeed related to the number of **real domains** and **deepfake methods**, rather than the number of real or fake images.

**Double-saturating power-law scaling is observed with respect to the number of training images.** To study the scaling law along this dimension, we randomly sample subsets of the ScaleDF training set at the following proportions: $1/4, 1/10, 1/16, 1/64, 1/100, 1/256, 1/1000$. Note that this differs from the previous section, *i.e.*, here we sample real and fake images simultaneously. To reduce randomness, each sampling is repeated 5 times. Each $\mu$ in Fig. 5 (Left Bottom) represents the mean computed over 5 repetitions and 7 test datasets, while $\sigma$ denotes the variance across the 5 repetitions. Based on these empirical data points, we observe that the trend of $\overline{\text{EER}}$ with respect to the number of training images ($N$) is best described by a double-saturating power law, which is the same form as originally proposed in image classification (Zhai et al., 2022), *i.e.*, $\overline{\text{EER}} = c + K \cdot (N + N_0)^{-\gamma}$. Using ordinary least squares (OLS), we estimate the parameters as $c = 0.175$, $K = 1.44 \cdot 10^3$, $N_0 = 1.19 \cdot 10^5$, and $\gamma = 0.710$. Meanwhile, we calculate the coefficient of determination $R^2 = 0.9989$, implying that the fitted power law explains $99.89\%$ of the variance in the observed data, thus providing an excellent description of the scaling relationship. This scaling law suggests that, with 46 real domains and 88 deepfake methods, performance gradually saturates once the total number of training images exceeds $10^7$. To further improve performance, blindly collecting more images from the same real domains or generating more from the same deepfake methods may **not be effective**. However, this does **not** imply that datasets larger than $10^7$ images offer no further benefit for deepfake detection. Rather, as we scale up the number of real domains and deepfake methods, more training data will still be essential.

**When training on ScaleDF, model size scaling appears to saturate at around 300 million parameters.** Beyond the aforementioned scaling laws, we are also interested in scaling along the model size dimension, *i.e.*, how large a model ScaleDF can support. Specifically, we select five different model sizes, denoted as ViT-S (21.7M), ViT-M (38.3M), ViT-B (85.8M), ViT-L (303.4M), and ViT-H (630.8M). Each $\mu$ in Fig. 5 (Right Bottom) represents the mean computed over 7 test datasets. Performance consistently improves from ViT-S (21.7M) to ViT-L (303.4M), but saturates thereafter, as further scaling to ViT-H (630.8M) yields no gains. This observation does **not** suggest that models with more than 300M parameters bring no additional gain for the deepfake detection task; rather, it indicates the maximum model size currently supported by the ScaleDF dataset. In the future, as we scale up real domains and deepfake methods, larger models may yield better performance.

## D    USED PERTURBATIONS

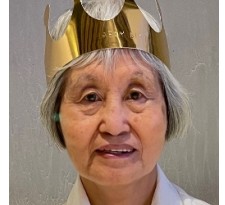

In this section, we describe how the perturbations from AnyPattern (Wang et al., 2024b) are utilized. AnyPattern is a large-scale perturbation dataset containing 100 perturbations. The code for generating each perturbation is available at here. However, since not all of them occur frequently in real-world scenarios, we select 30 common perturbations for training. To demonstrate the effect of the 30 selected perturbations, Fig. 6 shows the original face, while Tables 36, 37, and 38 present example faces with the applied perturbations. During training, for each image, we apply no perturbations with a probability of $50\%$, a single perturbation with a probability of $25\%$, and two perturbations with a probability of $25\%$.

Figure 6: Used original face.

## E    COMPLETE EXPERIMENTAL RESULTS FOR SCALING LAWS

In the main paper, we skip the exact values for each data point used in the scaling law analysis. To improve the reproducibility, we present all experimental results here in full detail. Specifically: (1) Tables 12 and 16 report the exact AUC and EER values for scaling laws with respect to the number of real domains; (2) Tables 13 and 17 report the exact AUC and EER values with respect to the number of deepfake methods; (3) Tables 14 and 18 report the exact AUC and EER values with respect to the number of training images; and (4) Tables 15 and 19 report the exact AUC and EER values with respect to different model sizes.

Table 8: Comparison of using different pre-training models: similar performance observed.

| EER | DFD | CDF V2 | Wild | Forgery. | DFF | DF40 | ScaleDF | Mean |
|---|---|---|---|---|---|---|---|---|
| ImageNet | 0.281 | 0.162 | 0.268 | 0.260 | 0.161 | 0.063 | 0.093 | 0.184 |
| CLIP | 0.278 | 0.179 | 0.279 | 0.231 | 0.085 | 0.065 | 0.052 | 0.167 |
| SigLIP 2 | 0.316 | 0.193 | 0.290 | 0.229 | 0.094 | 0.052 | 0.052 | 0.175 |

Table 9: Effectiveness of image quality compression (QC) and perturbations (PT).

| EER | DFD | CDF V2 | Wild | Forgery. | DFF | DF40 | ScaleDF | Mean |
|---|---|---|---|---|---|---|---|---|
| N/A | 0.391 | 0.270 | 0.299 | 0.315 | 0.261 | 0.136 | 0.066 | 0.248 |
| QC | 0.302 | 0.164 | 0.278 | 0.300 | 0.130 | 0.065 | 0.076 | 0.188 |
| QC + PT | 0.281 | 0.162 | 0.268 | 0.260 | 0.161 | 0.063 | 0.093 | 0.184 |

Table 10: Comparison of cross-benchmark performance: with scaling, we achieve the best performance.

| | EER | DFD | CDF V2 | Wild | Forgery. | DFF | DF40 | ScaleDF |
|---|---|---|---|---|---|---|---|---|
| | DFD | − | 0.276 | 0.322 | 0.425 | 0.470 | 0.428 | 0.488 |
| | CDF V2 | 0.352 | − | 0.288 | 0.401 | 0.419 | 0.383 | 0.417 |
| | Wild | 0.396 | 0.309 | − | 0.462 | 0.517 | 0.451 | 0.462 |
| Training sets | Forgery. | 0.267 | 0.170 | 0.275 | − | 0.372 | 0.251 | 0.390 |
| | DFF | 0.438 | 0.454 | 0.470 | 0.445 | − | 0.376 | 0.407 |
| | DF40 | 0.439 | 0.267 | 0.354 | 0.375 | 0.421 | − | 0.352 |
| | ScaleDF | 0.281 | 0.162 | 0.268 | 0.260 | 0.161 | 0.063 | − |

Table 11: Comparison of whether includes FaceForensics++ (Rössler et al., 2019) in ScaleDF.

| EER | DFD | CDF V2 | Wild | Forgery. | DFF | DF40 | ScaleDF | Mean |
|---|---|---|---|---|---|---|---|---|
| w/o FF++ | 0.319 | 0.188 | 0.276 | 0.290 | 0.185 | 0.083 | 0.081 | 0.203 |
| w FF++ | 0.281 | 0.162 | 0.268 | 0.260 | 0.161 | 0.063 | 0.093 | 0.184 |

## F  TRAINING DETAILS

We report training details here, basically following the scaling law reproducibility checklist (Li et al., 2025). The generation of ScaleDF required about $20,000$ A100 GPU hours. Training is distributed across 8 NVIDIA A100 40GB NVLink GPUs and 128 AMD EPYC 7742 CPU cores. Each training run requires approximately 160 GPU hours. Before training, all images are resized to a resolution of $224 \times 224$ pixels. We use a batch size of $2,048$ and train for 20 epochs with class balancing. The AdamW optimizer (Loshchilov & Hutter, 2017) is used with a cosine-decay learning rate schedule, a peak learning rate of $10^{-5}$, and 5 warm-up epochs.

## G  ADDITIONAL OBSERVATIONS IN EER

As shown in Tables 8 to 11, the observations in Section 5 still hold when evaluated with an alternative metric.

## H  ETHICS STATEMENT

We adhere to the license terms of all data and models in constructing ScaleDF. We are committed to responsible research, though there are open issues around fairness and bias. We put some important considerations here for interested readers.

### H.1  LICENSES

Our work builds upon numerous publicly available resources. In constructing ScaleDF, we have made every effort to comply with the licenses of all constituent datasets and deepfake generation methods. Full lists of these resources, along with direct links to their original sources, are provided in Table 6 and Table 7 to ensure transparency and enable reproducibility.

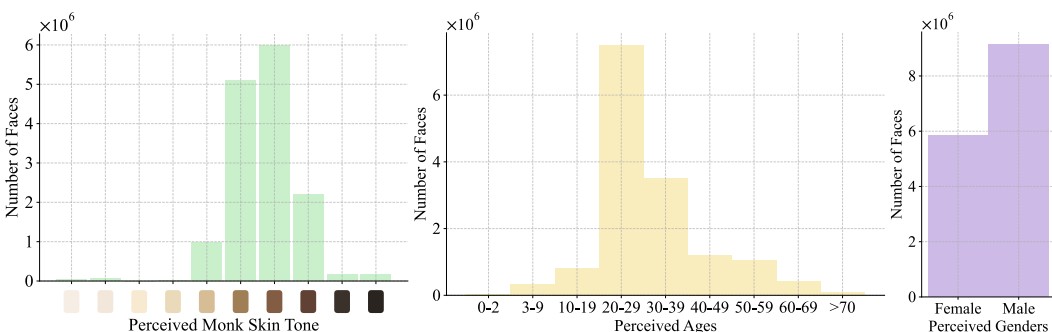

Figure 7: Perceived demographic distribution of faces in ScaleDF.

## H.2 FAIRNESS

Recognizing that large-scale datasets can inherit and amplify societal biases, we have analyzed the demographic distribution of what the faces appear to be within ScaleDF, using pre-trained models from third parties. While we aimed for inclusiveness, as shown in Fig. 7, our dataset inherits imbalances from the included real domains and deepfake methods, which we document here to ensure transparency and guide future research.

- **Perceived Monk Skin Tone (MST) Distribution.** The dataset shows an imbalance across different MSTs, with a strong concentration around MSTs 5, 6, 7, 8 (out of 10). This over-representation may result in performance disparities in detection models.
- **Perceived Age Distribution.** There is a strong concentration in the $20 \sim 29$ (about 7.5 million) and $30 \sim 39$ (about 3.5 million) perceived age groups. Younger and older individuals are under-represented, with groups like $> 70$ and $0 \sim 2$ containing fewer than 0.1 million faces each. This may limit the reliability of detectors for very young or elderly subjects.
- **Perceived Gender Distribution.** The perceived gender distribution is more balanced compared with age, but still shows a skew. The dataset includes about 8.8 million male faces and 5.8 million female faces.

We acknowledge these demographic biases as a limitation of ScaleDF. This is the nature of the best-effort data acquisition process from the public domain. We encourage future researchers to apply sampling techniques to promote a better fairness outcome.

## H.3 BROADER IMPACT

The primary goal of this research is to advance the capabilities of deepfake detection to combat malicious activities such as the spread of misinformation and the violation of personal privacy. By establishing predictable scaling laws, we provide the research community with valuable knowledge to build more robust and generalizable detectors, thereby strengthening societal defenses against manipulated media.

## I  LLM USAGE

We use an LLM-based writing assistant for minor grammar and style edits. All technical content, analyses, and conclusions are authored and verified by the human authors.

Table 12: Full experimental results measured by AUC for scaling law with varying numbers of *real domains*. The concluded scaling law is shown in Fig. 3 (Left).

| AUC | DFD | CDF V2 | Wild | Forgery. | DFF | DF40 | ScaleDF | Mean |
|-----|-----|--------|------|----------|-----|------|---------|------|
| 46 | 0.793 | 0.915 | 0.815 | 0.824 | 0.909 | 0.980 | 0.968 | 0.886 |
| 32 | 0.752 | 0.845 | 0.716 | 0.755 | 0.881 | 0.927 | 0.966 | 0.835 |
| 32 | 0.764 | 0.917 | 0.747 | 0.818 | 0.907 | 0.985 | 0.968 | 0.872 |
| 32 | 0.779 | 0.904 | 0.727 | 0.817 | 0.920 | 0.981 | 0.967 | 0.871 |
| 32 | 0.765 | 0.897 | 0.817 | 0.817 | 0.865 | 0.977 | 0.960 | 0.871 |
| 32 | 0.760 | 0.904 | 0.732 | 0.811 | 0.863 | 0.980 | 0.960 | 0.859 |
| 32 | 0.780 | 0.912 | 0.728 | 0.799 | 0.913 | 0.977 | 0.963 | 0.868 |
| 32 | 0.710 | 0.847 | 0.804 | 0.726 | 0.840 | 0.816 | 0.911 | 0.808 |
| 32 | 0.768 | 0.903 | 0.829 | 0.790 | 0.903 | 0.972 | 0.956 | 0.874 |
| 32 | 0.737 | 0.834 | 0.805 | 0.786 | 0.876 | 0.919 | 0.963 | 0.846 |
| 32 | 0.762 | 0.898 | 0.813 | 0.781 | 0.906 | 0.937 | 0.963 | 0.866 |
| 23 | 0.748 | 0.890 | 0.778 | 0.808 | 0.843 | 0.946 | 0.935 | 0.850 |
| 23 | 0.723 | 0.824 | 0.793 | 0.727 | 0.877 | 0.890 | 0.933 | 0.824 |
| 23 | 0.760 | 0.883 | 0.753 | 0.761 | 0.863 | 0.944 | 0.960 | 0.846 |
| 23 | 0.742 | 0.905 | 0.707 | 0.762 | 0.864 | 0.976 | 0.942 | 0.843 |
| 23 | 0.713 | 0.852 | 0.760 | 0.752 | 0.803 | 0.936 | 0.944 | 0.823 |
| 23 | 0.766 | 0.889 | 0.780 | 0.782 | 0.828 | 0.972 | 0.950 | 0.852 |
| 23 | 0.787 | 0.886 | 0.794 | 0.758 | 0.895 | 0.967 | 0.961 | 0.864 |
| 23 | 0.743 | 0.881 | 0.804 | 0.815 | 0.830 | 0.980 | 0.950 | 0.858 |
| 23 | 0.706 | 0.806 | 0.824 | 0.745 | 0.843 | 0.861 | 0.916 | 0.814 |
| 23 | 0.745 | 0.879 | 0.679 | 0.781 | 0.852 | 0.942 | 0.928 | 0.829 |
| 16 | 0.734 | 0.847 | 0.665 | 0.746 | 0.872 | 0.926 | 0.957 | 0.821 |
| 16 | 0.716 | 0.759 | 0.659 | 0.669 | 0.878 | 0.830 | 0.909 | 0.774 |
| 16 | 0.756 | 0.879 | 0.658 | 0.749 | 0.881 | 0.972 | 0.964 | 0.837 |
| 16 | 0.758 | 0.889 | 0.722 | 0.781 | 0.899 | 0.977 | 0.965 | 0.856 |
| 16 | 0.733 | 0.869 | 0.709 | 0.752 | 0.891 | 0.942 | 0.963 | 0.837 |
| 16 | 0.769 | 0.867 | 0.772 | 0.760 | 0.899 | 0.957 | 0.956 | 0.854 |
| 16 | 0.717 | 0.834 | 0.772 | 0.669 | 0.853 | 0.830 | 0.904 | 0.797 |
| 16 | 0.622 | 0.556 | 0.764 | 0.582 | 0.762 | 0.670 | 0.778 | 0.676 |
| 16 | 0.729 | 0.875 | 0.722 | 0.736 | 0.888 | 0.954 | 0.964 | 0.838 |
| 16 | 0.639 | 0.594 | 0.666 | 0.567 | 0.776 | 0.758 | 0.801 | 0.686 |
| 11 | 0.663 | 0.681 | 0.656 | 0.583 | 0.808 | 0.751 | 0.814 | 0.708 |
| 11 | 0.694 | 0.851 | 0.748 | 0.724 | 0.763 | 0.868 | 0.929 | 0.797 |
| 11 | 0.720 | 0.862 | 0.746 | 0.726 | 0.850 | 0.890 | 0.945 | 0.820 |
| 11 | 0.722 | 0.767 | 0.687 | 0.668 | 0.869 | 0.782 | 0.898 | 0.770 |
| 11 | 0.696 | 0.777 | 0.719 | 0.732 | 0.760 | 0.832 | 0.901 | 0.774 |
| 11 | 0.714 | 0.806 | 0.644 | 0.638 | 0.822 | 0.796 | 0.854 | 0.754 |
| 11 | 0.654 | 0.767 | 0.619 | 0.627 | 0.818 | 0.742 | 0.822 | 0.721 |
| 11 | 0.715 | 0.823 | 0.738 | 0.721 | 0.813 | 0.936 | 0.949 | 0.814 |
| 11 | 0.680 | 0.712 | 0.721 | 0.579 | 0.852 | 0.744 | 0.791 | 0.726 |
| 11 | 0.713 | 0.756 | 0.753 | 0.708 | 0.756 | 0.858 | 0.835 | 0.768 |
| 8 | 0.659 | 0.623 | 0.615 | 0.588 | 0.783 | 0.695 | 0.796 | 0.680 |
| 8 | 0.550 | 0.721 | 0.608 | 0.555 | 0.599 | 0.692 | 0.692 | 0.631 |
| 8 | 0.716 | 0.745 | 0.622 | 0.698 | 0.848 | 0.894 | 0.875 | 0.771 |
| 8 | 0.678 | 0.794 | 0.649 | 0.704 | 0.664 | 0.946 | 0.926 | 0.766 |
| 8 | 0.674 | 0.607 | 0.617 | 0.557 | 0.890 | 0.735 | 0.798 | 0.697 |
| 8 | 0.667 | 0.689 | 0.626 | 0.631 | 0.747 | 0.684 | 0.806 | 0.693 |
| 8 | 0.673 | 0.730 | 0.725 | 0.688 | 0.781 | 0.825 | 0.885 | 0.758 |
| 8 | 0.690 | 0.785 | 0.546 | 0.671 | 0.747 | 0.918 | 0.837 | 0.742 |
| 8 | 0.690 | 0.795 | 0.581 | 0.651 | 0.877 | 0.730 | 0.827 | 0.736 |
| 8 | 0.524 | 0.480 | 0.688 | 0.488 | 0.595 | 0.723 | 0.726 | 0.603 |
| 5 | 0.586 | 0.441 | 0.636 | 0.533 | 0.758 | 0.662 | 0.739 | 0.622 |
| 5 | 0.610 | 0.518 | 0.590 | 0.542 | 0.788 | 0.700 | 0.765 | 0.645 |
| 5 | 0.663 | 0.686 | 0.682 | 0.598 | 0.854 | 0.785 | 0.832 | 0.728 |
| 5 | 0.592 | 0.666 | 0.718 | 0.591 | 0.643 | 0.857 | 0.770 | 0.691 |
| 5 | 0.689 | 0.771 | 0.658 | 0.685 | 0.673 | 0.853 | 0.689 | 0.717 |
| 5 | 0.548 | 0.475 | 0.566 | 0.542 | 0.744 | 0.652 | 0.624 | 0.593 |
| 5 | 0.627 | 0.561 | 0.633 | 0.563 | 0.750 | 0.659 | 0.645 | 0.634 |
| 5 | 0.696 | 0.811 | 0.664 | 0.696 | 0.802 | 0.814 | 0.834 | 0.760 |
| 5 | 0.676 | 0.686 | 0.690 | 0.615 | 0.651 | 0.764 | 0.822 | 0.701 |
| 5 | 0.713 | 0.758 | 0.565 | 0.671 | 0.862 | 0.898 | 0.890 | 0.765 |

Table 13: Full experimental results measured by AUC for scaling law with varying numbers of *deepfake methods*. The concluded scaling law is shown in Fig. 3 (Right).

| AUC | DFD | CDF V2 | Wild | Forgery. | DFF | DF40 | ScaleDF | Mean |
|---|---|---|---|---|---|---|---|---|
| 88 | 0.793 | 0.915 | 0.815 | 0.824 | 0.909 | 0.980 | 0.968 | 0.886 |
| 64 | 0.774 | 0.914 | 0.776 | 0.815 | 0.874 | 0.982 | 0.965 | 0.871 |
| 64 | 0.756 | 0.912 | 0.802 | 0.820 | 0.893 | 0.978 | 0.955 | 0.874 |
| 64 | 0.790 | 0.917 | 0.802 | 0.836 | 0.888 | 0.981 | 0.963 | 0.882 |
| 64 | 0.782 | 0.915 | 0.796 | 0.823 | 0.888 | 0.977 | 0.963 | 0.878 |
| 64 | 0.766 | 0.908 | 0.806 | 0.802 | 0.894 | 0.978 | 0.965 | 0.874 |
| 64 | 0.755 | 0.853 | 0.809 | 0.803 | 0.858 | 0.973 | 0.967 | 0.860 |
| 64 | 0.756 | 0.873 | 0.823 | 0.795 | 0.818 | 0.977 | 0.959 | 0.857 |
| 64 | 0.779 | 0.899 | 0.800 | 0.804 | 0.899 | 0.977 | 0.960 | 0.874 |
| 64 | 0.766 | 0.894 | 0.828 | 0.808 | 0.909 | 0.977 | 0.960 | 0.877 |
| 64 | 0.788 | 0.909 | 0.819 | 0.819 | 0.891 | 0.975 | 0.964 | 0.881 |
| 45 | 0.777 | 0.886 | 0.794 | 0.798 | 0.761 | 0.975 | 0.954 | 0.849 |
| 45 | 0.766 | 0.906 | 0.804 | 0.792 | 0.781 | 0.970 | 0.954 | 0.853 |
| 45 | 0.746 | 0.909 | 0.748 | 0.802 | 0.772 | 0.974 | 0.966 | 0.845 |
| 45 | 0.760 | 0.915 | 0.754 | 0.791 | 0.893 | 0.967 | 0.946 | 0.861 |
| 45 | 0.763 | 0.904 | 0.791 | 0.810 | 0.796 | 0.979 | 0.953 | 0.857 |
| 45 | 0.684 | 0.824 | 0.800 | 0.774 | 0.882 | 0.963 | 0.959 | 0.841 |
| 45 | 0.673 | 0.854 | 0.775 | 0.792 | 0.842 | 0.976 | 0.958 | 0.839 |
| 45 | 0.701 | 0.897 | 0.768 | 0.778 | 0.768 | 0.968 | 0.937 | 0.831 |
| 45 | 0.764 | 0.908 | 0.809 | 0.811 | 0.829 | 0.970 | 0.956 | 0.864 |
| 45 | 0.796 | 0.912 | 0.816 | 0.817 | 0.816 | 0.969 | 0.955 | 0.869 |
| 32 | 0.720 | 0.794 | 0.758 | 0.790 | 0.855 | 0.973 | 0.949 | 0.834 |
| 32 | 0.767 | 0.869 | 0.799 | 0.751 | 0.856 | 0.969 | 0.942 | 0.850 |
| 32 | 0.679 | 0.877 | 0.754 | 0.765 | 0.718 | 0.959 | 0.947 | 0.814 |
| 32 | 0.669 | 0.839 | 0.752 | 0.743 | 0.853 | 0.947 | 0.950 | 0.822 |
| 32 | 0.685 | 0.836 | 0.740 | 0.762 | 0.870 | 0.970 | 0.949 | 0.830 |
| 32 | 0.740 | 0.892 | 0.741 | 0.764 | 0.794 | 0.952 | 0.931 | 0.830 |
| 32 | 0.734 | 0.845 | 0.773 | 0.765 | 0.774 | 0.951 | 0.915 | 0.822 |
| 32 | 0.713 | 0.804 | 0.819 | 0.768 | 0.722 | 0.968 | 0.934 | 0.818 |
| 32 | 0.742 | 0.842 | 0.782 | 0.772 | 0.818 | 0.969 | 0.952 | 0.840 |
| 32 | 0.748 | 0.910 | 0.708 | 0.767 | 0.759 | 0.968 | 0.944 | 0.829 |
| 22 | 0.667 | 0.891 | 0.733 | 0.791 | 0.663 | 0.965 | 0.935 | 0.806 |
| 22 | 0.662 | 0.829 | 0.729 | 0.732 | 0.854 | 0.959 | 0.924 | 0.813 |
| 22 | 0.736 | 0.836 | 0.772 | 0.736 | 0.725 | 0.957 | 0.909 | 0.810 |
| 22 | 0.625 | 0.774 | 0.744 | 0.726 | 0.857 | 0.945 | 0.923 | 0.799 |
| 22 | 0.736 | 0.842 | 0.746 | 0.803 | 0.880 | 0.966 | 0.909 | 0.840 |
| 22 | 0.648 | 0.813 | 0.651 | 0.758 | 0.654 | 0.971 | 0.932 | 0.775 |
| 22 | 0.669 | 0.826 | 0.710 | 0.744 | 0.858 | 0.958 | 0.926 | 0.813 |
| 22 | 0.583 | 0.743 | 0.717 | 0.727 | 0.700 | 0.957 | 0.929 | 0.765 |
| 22 | 0.744 | 0.801 | 0.779 | 0.766 | 0.726 | 0.960 | 0.933 | 0.816 |
| 22 | 0.708 | 0.813 | 0.771 | 0.763 | 0.618 | 0.943 | 0.899 | 0.788 |
| 16 | 0.626 | 0.822 | 0.708 | 0.749 | 0.677 | 0.956 | 0.923 | 0.780 |
| 16 | 0.609 | 0.728 | 0.698 | 0.693 | 0.854 | 0.957 | 0.942 | 0.783 |
| 16 | 0.779 | 0.899 | 0.755 | 0.804 | 0.724 | 0.870 | 0.874 | 0.815 |
| 16 | 0.566 | 0.656 | 0.707 | 0.697 | 0.643 | 0.956 | 0.893 | 0.731 |
| 16 | 0.590 | 0.674 | 0.686 | 0.716 | 0.720 | 0.963 | 0.937 | 0.755 |
| 16 | 0.701 | 0.810 | 0.766 | 0.778 | 0.634 | 0.955 | 0.916 | 0.794 |
| 16 | 0.660 | 0.827 | 0.718 | 0.739 | 0.659 | 0.968 | 0.912 | 0.783 |
| 16 | 0.556 | 0.674 | 0.684 | 0.660 | 0.858 | 0.954 | 0.916 | 0.757 |
| 16 | 0.617 | 0.804 | 0.684 | 0.699 | 0.722 | 0.943 | 0.903 | 0.767 |
| 16 | 0.677 | 0.829 | 0.644 | 0.703 | 0.609 | 0.907 | 0.893 | 0.752 |
| 11 | 0.555 | 0.662 | 0.718 | 0.678 | 0.862 | 0.956 | 0.906 | 0.763 |
| 11 | 0.604 | 0.775 | 0.689 | 0.720 | 0.656 | 0.880 | 0.890 | 0.745 |
| 11 | 0.710 | 0.748 | 0.763 | 0.754 | 0.593 | 0.942 | 0.880 | 0.770 |
| 11 | 0.706 | 0.814 | 0.751 | 0.774 | 0.744 | 0.965 | 0.887 | 0.806 |
| 11 | 0.740 | 0.848 | 0.723 | 0.756 | 0.519 | 0.956 | 0.901 | 0.778 |
| 11 | 0.687 | 0.826 | 0.606 | 0.752 | 0.510 | 0.910 | 0.865 | 0.736 |
| 11 | 0.686 | 0.756 | 0.743 | 0.768 | 0.607 | 0.858 | 0.861 | 0.754 |
| 11 | 0.662 | 0.771 | 0.594 | 0.714 | 0.537 | 0.919 | 0.862 | 0.723 |
| 11 | 0.592 | 0.745 | 0.603 | 0.708 | 0.649 | 0.905 | 0.898 | 0.728 |
| 11 | 0.742 | 0.806 | 0.743 | 0.746 | 0.795 | 0.906 | 0.856 | 0.799 |

Table 14: Full experimental results measured by AUC for scaling law with varying numbers of *training images*. The concluded scaling law is shown in Fig. 4 (Left).

| AUC | DFD | CDF V2 | Wild | Forgery. | DFF | DF40 | ScaleDF | Mean |
|---|---|---|---|---|---|---|---|---|
| $1.4 \times 10^7$ | 0.793 | 0.915 | 0.815 | 0.824 | 0.909 | 0.980 | 0.968 | 0.886 |
| $3.7 \times 10^6$ | 0.788 | 0.904 | 0.831 | 0.792 | 0.842 | 0.964 | 0.949 | 0.867 |
| $3.7 \times 10^6$ | 0.778 | 0.903 | 0.827 | 0.790 | 0.840 | 0.966 | 0.948 | 0.864 |
| $3.7 \times 10^6$ | 0.781 | 0.905 | 0.820 | 0.791 | 0.840 | 0.962 | 0.946 | 0.864 |
| $3.7 \times 10^6$ | 0.777 | 0.899 | 0.823 | 0.790 | 0.843 | 0.967 | 0.950 | 0.864 |
| $3.7 \times 10^6$ | 0.785 | 0.904 | 0.820 | 0.791 | 0.838 | 0.963 | 0.946 | 0.864 |
| $1.4 \times 10^6$ | 0.754 | 0.873 | 0.828 | 0.756 | 0.804 | 0.952 | 0.943 | 0.844 |
| $1.4 \times 10^6$ | 0.763 | 0.876 | 0.828 | 0.758 | 0.801 | 0.949 | 0.941 | 0.845 |
| $1.4 \times 10^6$ | 0.761 | 0.880 | 0.824 | 0.757 | 0.805 | 0.948 | 0.942 | 0.845 |
| $1.4 \times 10^6$ | 0.760 | 0.878 | 0.823 | 0.756 | 0.812 | 0.948 | 0.940 | 0.845 |
| $1.4 \times 10^6$ | 0.760 | 0.867 | 0.823 | 0.756 | 0.811 | 0.950 | 0.939 | 0.844 |
| $9.3 \times 10^5$ | 0.725 | 0.850 | 0.805 | 0.724 | 0.776 | 0.941 | 0.933 | 0.822 |
| $9.3 \times 10^5$ | 0.734 | 0.845 | 0.803 | 0.727 | 0.780 | 0.943 | 0.933 | 0.824 |
| $9.3 \times 10^5$ | 0.735 | 0.850 | 0.804 | 0.723 | 0.781 | 0.939 | 0.932 | 0.824 |
| $9.3 \times 10^5$ | 0.726 | 0.849 | 0.801 | 0.722 | 0.776 | 0.936 | 0.928 | 0.820 |
| $9.3 \times 10^5$ | 0.737 | 0.854 | 0.803 | 0.726 | 0.778 | 0.941 | 0.932 | 0.824 |
| $2.3 \times 10^5$ | 0.624 | 0.668 | 0.718 | 0.614 | 0.708 | 0.830 | 0.848 | 0.716 |
| $2.3 \times 10^5$ | 0.628 | 0.676 | 0.716 | 0.616 | 0.714 | 0.843 | 0.855 | 0.721 |
| $2.3 \times 10^5$ | 0.630 | 0.673 | 0.724 | 0.617 | 0.713 | 0.826 | 0.850 | 0.719 |
| $2.3 \times 10^5$ | 0.628 | 0.679 | 0.721 | 0.616 | 0.710 | 0.831 | 0.851 | 0.719 |
| $2.3 \times 10^5$ | 0.614 | 0.669 | 0.727 | 0.615 | 0.713 | 0.830 | 0.853 | 0.717 |
| $1.4 \times 10^5$ | 0.608 | 0.654 | 0.699 | 0.596 | 0.706 | 0.744 | 0.786 | 0.685 |
| $1.4 \times 10^5$ | 0.602 | 0.648 | 0.680 | 0.597 | 0.696 | 0.735 | 0.782 | 0.677 |
| $1.4 \times 10^5$ | 0.605 | 0.644 | 0.698 | 0.596 | 0.699 | 0.732 | 0.784 | 0.680 |
| $1.4 \times 10^5$ | 0.604 | 0.655 | 0.696 | 0.599 | 0.706 | 0.731 | 0.782 | 0.682 |
| $1.4 \times 10^5$ | 0.604 | 0.645 | 0.679 | 0.598 | 0.697 | 0.734 | 0.787 | 0.678 |
| $5.8 \times 10^4$ | 0.551 | 0.536 | 0.570 | 0.554 | 0.652 | 0.464 | 0.656 | 0.569 |
| $5.8 \times 10^4$ | 0.550 | 0.545 | 0.573 | 0.555 | 0.657 | 0.470 | 0.661 | 0.573 |
| $5.8 \times 10^4$ | 0.558 | 0.542 | 0.565 | 0.556 | 0.656 | 0.474 | 0.659 | 0.573 |
| $5.8 \times 10^4$ | 0.553 | 0.548 | 0.579 | 0.555 | 0.651 | 0.464 | 0.658 | 0.573 |
| $5.8 \times 10^4$ | 0.552 | 0.542 | 0.561 | 0.550 | 0.644 | 0.450 | 0.654 | 0.565 |
| $1.4 \times 10^4$ | 0.509 | 0.509 | 0.530 | 0.511 | 0.580 | 0.344 | 0.548 | 0.504 |
| $1.4 \times 10^4$ | 0.507 | 0.491 | 0.539 | 0.511 | 0.581 | 0.335 | 0.550 | 0.502 |
| $1.4 \times 10^4$ | 0.506 | 0.492 | 0.536 | 0.513 | 0.581 | 0.353 | 0.555 | 0.505 |
| $1.4 \times 10^4$ | 0.506 | 0.501 | 0.527 | 0.511 | 0.583 | 0.350 | 0.555 | 0.505 |
| $1.4 \times 10^4$ | 0.508 | 0.489 | 0.535 | 0.511 | 0.586 | 0.339 | 0.551 | 0.503 |

Table 15: Full experimental results measured by AUC for varying *model sizes*. The average performance is shown in Fig. 4 (Right).

| AUC | DFD | CDF V2 | Wild | Forgery. | DFF | DF40 | ScaleDF | Mean |
|---|---|---|---|---|---|---|---|---|
| 21.7M | 0.785 | 0.905 | 0.807 | 0.774 | 0.807 | 0.970 | 0.948 | 0.856 |
| 38.3M | 0.764 | 0.905 | 0.812 | 0.795 | 0.843 | 0.974 | 0.953 | 0.864 |
| 85.8M | 0.793 | 0.915 | 0.815 | 0.824 | 0.909 | 0.980 | 0.968 | 0.886 |
| 303.4M | 0.803 | 0.925 | 0.808 | 0.859 | 0.940 | 0.985 | 0.976 | 0.900 |
| 630.8M | 0.795 | 0.911 | 0.797 | 0.867 | 0.959 | 0.985 | 0.982 | 0.899 |

Table 16: Full experimental results measured by EER for scaling law with varying numbers of *real domains*. The concluded scaling law is shown in Fig. 5 (Left Top).

| EER | DFD | CDF V2 | Wild | Forgery. | DFF | DF40 | ScaleDF | Mean |
|---|---|---|---|---|---|---|---|---|
| 46 | 0.281 | 0.162 | 0.268 | 0.260 | 0.161 | 0.063 | 0.093 | 0.184 |
| 32 | 0.317 | 0.224 | 0.337 | 0.310 | 0.189 | 0.149 | 0.092 | 0.231 |
| 32 | 0.308 | 0.162 | 0.313 | 0.264 | 0.162 | 0.055 | 0.092 | 0.194 |
| 32 | 0.293 | 0.173 | 0.326 | 0.265 | 0.148 | 0.063 | 0.093 | 0.194 |
| 32 | 0.307 | 0.179 | 0.257 | 0.269 | 0.209 | 0.070 | 0.105 | 0.199 |
| 32 | 0.308 | 0.173 | 0.334 | 0.270 | 0.210 | 0.064 | 0.104 | 0.209 |
| 32 | 0.291 | 0.165 | 0.336 | 0.275 | 0.158 | 0.069 | 0.098 | 0.199 |
| 32 | 0.344 | 0.221 | 0.262 | 0.336 | 0.229 | 0.250 | 0.152 | 0.256 |
| 32 | 0.295 | 0.170 | 0.246 | 0.287 | 0.166 | 0.078 | 0.107 | 0.193 |
| 32 | 0.330 | 0.240 | 0.268 | 0.297 | 0.197 | 0.158 | 0.102 | 0.228 |
| 32 | 0.309 | 0.183 | 0.261 | 0.287 | 0.168 | 0.134 | 0.102 | 0.206 |
| 23 | 0.316 | 0.192 | 0.285 | 0.276 | 0.227 | 0.127 | 0.135 | 0.223 |
| 23 | 0.331 | 0.250 | 0.288 | 0.334 | 0.197 | 0.194 | 0.146 | 0.249 |
| 23 | 0.312 | 0.196 | 0.314 | 0.306 | 0.214 | 0.125 | 0.105 | 0.225 |
| 23 | 0.320 | 0.174 | 0.335 | 0.302 | 0.209 | 0.069 | 0.127 | 0.220 |
| 23 | 0.351 | 0.225 | 0.310 | 0.316 | 0.267 | 0.136 | 0.130 | 0.248 |
| 23 | 0.304 | 0.190 | 0.281 | 0.294 | 0.246 | 0.075 | 0.123 | 0.216 |
| 23 | 0.285 | 0.189 | 0.268 | 0.312 | 0.176 | 0.081 | 0.096 | 0.201 |
| 23 | 0.327 | 0.196 | 0.272 | 0.271 | 0.243 | 0.065 | 0.127 | 0.215 |
| 23 | 0.343 | 0.273 | 0.256 | 0.327 | 0.227 | 0.215 | 0.170 | 0.259 |
| 23 | 0.319 | 0.199 | 0.374 | 0.295 | 0.228 | 0.131 | 0.153 | 0.243 |
| 16 | 0.335 | 0.226 | 0.355 | 0.319 | 0.202 | 0.156 | 0.109 | 0.243 |
| 16 | 0.342 | 0.308 | 0.373 | 0.378 | 0.189 | 0.235 | 0.165 | 0.284 |
| 16 | 0.320 | 0.200 | 0.393 | 0.311 | 0.190 | 0.083 | 0.091 | 0.227 |
| 16 | 0.311 | 0.188 | 0.335 | 0.292 | 0.172 | 0.064 | 0.089 | 0.208 |
| 16 | 0.333 | 0.205 | 0.334 | 0.314 | 0.179 | 0.129 | 0.093 | 0.227 |
| 16 | 0.299 | 0.198 | 0.293 | 0.309 | 0.172 | 0.097 | 0.093 | 0.209 |
| 16 | 0.342 | 0.238 | 0.295 | 0.376 | 0.223 | 0.250 | 0.169 | 0.270 |
| 16 | 0.415 | 0.457 | 0.287 | 0.444 | 0.295 | 0.358 | 0.312 | 0.367 |
| 16 | 0.337 | 0.201 | 0.335 | 0.326 | 0.182 | 0.110 | 0.092 | 0.226 |
| 16 | 0.404 | 0.436 | 0.388 | 0.460 | 0.286 | 0.293 | 0.302 | 0.367 |
| 11 | 0.387 | 0.374 | 0.387 | 0.447 | 0.256 | 0.297 | 0.282 | 0.347 |
| 11 | 0.358 | 0.229 | 0.310 | 0.335 | 0.300 | 0.198 | 0.140 | 0.267 |
| 11 | 0.331 | 0.218 | 0.308 | 0.333 | 0.222 | 0.191 | 0.119 | 0.246 |
| 11 | 0.337 | 0.293 | 0.369 | 0.377 | 0.201 | 0.284 | 0.158 | 0.288 |
| 11 | 0.347 | 0.294 | 0.318 | 0.335 | 0.299 | 0.241 | 0.170 | 0.286 |
| 11 | 0.337 | 0.257 | 0.402 | 0.397 | 0.252 | 0.280 | 0.208 | 0.305 |
| 11 | 0.396 | 0.295 | 0.421 | 0.412 | 0.255 | 0.324 | 0.254 | 0.337 |
| 11 | 0.351 | 0.250 | 0.316 | 0.337 | 0.255 | 0.143 | 0.118 | 0.253 |
| 11 | 0.374 | 0.344 | 0.323 | 0.445 | 0.218 | 0.313 | 0.286 | 0.329 |
| 11 | 0.339 | 0.312 | 0.315 | 0.354 | 0.311 | 0.224 | 0.237 | 0.299 |
| 8 | 0.383 | 0.421 | 0.421 | 0.442 | 0.292 | 0.364 | 0.251 | 0.367 |
| 8 | 0.469 | 0.337 | 0.410 | 0.464 | 0.422 | 0.359 | 0.360 | 0.403 |
| 8 | 0.348 | 0.314 | 0.402 | 0.353 | 0.222 | 0.185 | 0.180 | 0.286 |
| 8 | 0.378 | 0.278 | 0.402 | 0.351 | 0.384 | 0.122 | 0.150 | 0.295 |
| 8 | 0.376 | 0.428 | 0.406 | 0.465 | 0.177 | 0.317 | 0.294 | 0.352 |
| 8 | 0.379 | 0.369 | 0.385 | 0.406 | 0.308 | 0.354 | 0.268 | 0.353 |
| 8 | 0.372 | 0.336 | 0.324 | 0.367 | 0.282 | 0.247 | 0.194 | 0.303 |
| 8 | 0.370 | 0.276 | 0.464 | 0.376 | 0.312 | 0.155 | 0.214 | 0.310 |
| 8 | 0.358 | 0.270 | 0.422 | 0.393 | 0.195 | 0.326 | 0.222 | 0.312 |
| 8 | 0.484 | 0.514 | 0.354 | 0.512 | 0.427 | 0.333 | 0.342 | 0.424 |
| 5 | 0.438 | 0.542 | 0.397 | 0.481 | 0.294 | 0.373 | 0.324 | 0.407 |
| 5 | 0.426 | 0.486 | 0.431 | 0.476 | 0.267 | 0.344 | 0.311 | 0.392 |
| 5 | 0.383 | 0.374 | 0.363 | 0.435 | 0.220 | 0.275 | 0.260 | 0.330 |
| 5 | 0.435 | 0.384 | 0.337 | 0.439 | 0.390 | 0.217 | 0.302 | 0.358 |
| 5 | 0.359 | 0.300 | 0.390 | 0.371 | 0.366 | 0.227 | 0.369 | 0.340 |
| 5 | 0.471 | 0.516 | 0.448 | 0.481 | 0.326 | 0.382 | 0.413 | 0.434 |
| 5 | 0.408 | 0.466 | 0.389 | 0.462 | 0.310 | 0.385 | 0.388 | 0.401 |
| 5 | 0.356 | 0.260 | 0.380 | 0.356 | 0.265 | 0.253 | 0.242 | 0.302 |
| 5 | 0.368 | 0.369 | 0.361 | 0.422 | 0.396 | 0.299 | 0.264 | 0.354 |
| 5 | 0.339 | 0.309 | 0.457 | 0.371 | 0.206 | 0.176 | 0.159 | 0.288 |

Table 17: Full experimental results measured by EER for scaling law with varying numbers of *deepfake methods*. The concluded scaling law is shown in Fig. 5 (Right Top).

| EER | DFD | CDF V2 | Wild | Forgery. | DFF | DF40 | ScaleDF | Mean |
|---|---|---|---|---|---|---|---|---|
| 88 | 0.281 | 0.162 | 0.268 | 0.260 | 0.161 | 0.063 | 0.093 | 0.184 |
| 64 | 0.299 | 0.165 | 0.301 | 0.268 | 0.200 | 0.061 | 0.096 | 0.199 |
| 64 | 0.312 | 0.167 | 0.279 | 0.264 | 0.175 | 0.066 | 0.112 | 0.196 |
| 64 | 0.283 | 0.162 | 0.277 | 0.249 | 0.186 | 0.064 | 0.099 | 0.189 |
| 64 | 0.293 | 0.162 | 0.272 | 0.261 | 0.183 | 0.067 | 0.100 | 0.191 |
| 64 | 0.304 | 0.172 | 0.278 | 0.278 | 0.178 | 0.068 | 0.096 | 0.196 |
| 64 | 0.310 | 0.222 | 0.266 | 0.278 | 0.216 | 0.077 | 0.094 | 0.209 |
| 64 | 0.314 | 0.208 | 0.255 | 0.286 | 0.253 | 0.071 | 0.106 | 0.213 |
| 64 | 0.294 | 0.180 | 0.274 | 0.278 | 0.173 | 0.070 | 0.107 | 0.197 |
| 64 | 0.303 | 0.183 | 0.251 | 0.273 | 0.162 | 0.070 | 0.106 | 0.193 |
| 64 | 0.287 | 0.169 | 0.257 | 0.264 | 0.182 | 0.073 | 0.097 | 0.190 |
| 45 | 0.298 | 0.197 | 0.276 | 0.282 | 0.304 | 0.073 | 0.113 | 0.221 |
| 45 | 0.306 | 0.171 | 0.270 | 0.286 | 0.291 | 0.080 | 0.118 | 0.217 |
| 45 | 0.318 | 0.171 | 0.309 | 0.277 | 0.289 | 0.077 | 0.092 | 0.219 |
| 45 | 0.307 | 0.159 | 0.309 | 0.287 | 0.180 | 0.088 | 0.121 | 0.207 |
| 45 | 0.307 | 0.177 | 0.285 | 0.270 | 0.269 | 0.067 | 0.114 | 0.213 |
| 45 | 0.372 | 0.251 | 0.275 | 0.301 | 0.194 | 0.095 | 0.107 | 0.228 |
| 45 | 0.376 | 0.228 | 0.292 | 0.286 | 0.230 | 0.075 | 0.109 | 0.228 |
| 45 | 0.363 | 0.185 | 0.301 | 0.298 | 0.293 | 0.086 | 0.139 | 0.238 |
| 45 | 0.307 | 0.172 | 0.261 | 0.272 | 0.243 | 0.085 | 0.110 | 0.207 |
| 45 | 0.280 | 0.164 | 0.265 | 0.268 | 0.262 | 0.082 | 0.114 | 0.205 |
| 32 | 0.340 | 0.279 | 0.311 | 0.289 | 0.220 | 0.081 | 0.122 | 0.235 |
| 32 | 0.312 | 0.216 | 0.274 | 0.323 | 0.221 | 0.085 | 0.129 | 0.223 |
| 32 | 0.372 | 0.205 | 0.309 | 0.309 | 0.340 | 0.096 | 0.122 | 0.250 |
| 32 | 0.379 | 0.240 | 0.307 | 0.326 | 0.222 | 0.117 | 0.123 | 0.245 |
| 32 | 0.370 | 0.241 | 0.315 | 0.313 | 0.210 | 0.085 | 0.122 | 0.237 |
| 32 | 0.326 | 0.188 | 0.317 | 0.310 | 0.282 | 0.114 | 0.145 | 0.240 |
| 32 | 0.329 | 0.233 | 0.294 | 0.309 | 0.297 | 0.112 | 0.164 | 0.248 |
| 32 | 0.345 | 0.272 | 0.256 | 0.306 | 0.337 | 0.092 | 0.146 | 0.251 |
| 32 | 0.321 | 0.241 | 0.293 | 0.303 | 0.260 | 0.085 | 0.123 | 0.232 |
| 32 | 0.317 | 0.168 | 0.344 | 0.308 | 0.308 | 0.089 | 0.130 | 0.238 |
| 22 | 0.390 | 0.194 | 0.330 | 0.282 | 0.382 | 0.090 | 0.130 | 0.257 |
| 22 | 0.388 | 0.251 | 0.325 | 0.335 | 0.222 | 0.101 | 0.148 | 0.253 |
| 22 | 0.326 | 0.244 | 0.298 | 0.332 | 0.332 | 0.100 | 0.168 | 0.257 |
| 22 | 0.413 | 0.300 | 0.311 | 0.338 | 0.218 | 0.125 | 0.148 | 0.265 |
| 22 | 0.329 | 0.237 | 0.317 | 0.278 | 0.192 | 0.088 | 0.167 | 0.230 |
| 22 | 0.397 | 0.265 | 0.375 | 0.313 | 0.395 | 0.081 | 0.135 | 0.280 |
| 22 | 0.386 | 0.252 | 0.344 | 0.328 | 0.216 | 0.102 | 0.155 | 0.255 |
| 22 | 0.443 | 0.323 | 0.333 | 0.336 | 0.354 | 0.104 | 0.141 | 0.291 |
| 22 | 0.325 | 0.276 | 0.288 | 0.310 | 0.334 | 0.105 | 0.147 | 0.255 |
| 22 | 0.344 | 0.262 | 0.299 | 0.310 | 0.420 | 0.120 | 0.174 | 0.275 |
| 16 | 0.407 | 0.254 | 0.345 | 0.315 | 0.372 | 0.106 | 0.149 | 0.278 |
| 16 | 0.428 | 0.331 | 0.347 | 0.364 | 0.226 | 0.109 | 0.132 | 0.277 |
| 16 | 0.294 | 0.181 | 0.309 | 0.277 | 0.331 | 0.190 | 0.202 | 0.255 |
| 16 | 0.454 | 0.385 | 0.352 | 0.355 | 0.405 | 0.101 | 0.172 | 0.318 |
| 16 | 0.437 | 0.373 | 0.357 | 0.345 | 0.338 | 0.092 | 0.126 | 0.296 |
| 16 | 0.355 | 0.268 | 0.294 | 0.298 | 0.420 | 0.109 | 0.158 | 0.272 |
| 16 | 0.396 | 0.250 | 0.331 | 0.326 | 0.389 | 0.080 | 0.166 | 0.277 |
| 16 | 0.466 | 0.374 | 0.354 | 0.388 | 0.218 | 0.115 | 0.163 | 0.297 |
| 16 | 0.416 | 0.273 | 0.366 | 0.354 | 0.341 | 0.122 | 0.168 | 0.291 |
| 16 | 0.374 | 0.246 | 0.390 | 0.358 | 0.421 | 0.157 | 0.199 | 0.307 |
| 11 | 0.465 | 0.386 | 0.334 | 0.374 | 0.216 | 0.111 | 0.177 | 0.295 |
| 11 | 0.426 | 0.295 | 0.366 | 0.346 | 0.385 | 0.210 | 0.198 | 0.318 |
| 11 | 0.352 | 0.319 | 0.302 | 0.315 | 0.440 | 0.124 | 0.196 | 0.293 |
| 11 | 0.353 | 0.266 | 0.310 | 0.301 | 0.325 | 0.091 | 0.182 | 0.261 |
| 11 | 0.328 | 0.234 | 0.335 | 0.313 | 0.493 | 0.106 | 0.172 | 0.283 |
| 11 | 0.367 | 0.252 | 0.420 | 0.317 | 0.501 | 0.158 | 0.215 | 0.318 |
| 11 | 0.363 | 0.310 | 0.323 | 0.304 | 0.439 | 0.191 | 0.210 | 0.305 |
| 11 | 0.382 | 0.297 | 0.435 | 0.342 | 0.481 | 0.151 | 0.210 | 0.328 |
| 11 | 0.436 | 0.323 | 0.434 | 0.346 | 0.402 | 0.165 | 0.169 | 0.325 |
| 11 | 0.321 | 0.271 | 0.332 | 0.323 | 0.287 | 0.160 | 0.217 | 0.273 |

Table 18: Full experimental results measured by EER for scaling law with varying numbers of *training images*. The concluded scaling law is shown in Fig. 5 (Left Bottom).

| EER | DFD | CDF V2 | Wild | Forgery. | DFF | DF40 | ScaleDF | Mean |
|---|---|---|---|---|---|---|---|---|
| $1.4 \times 10^7$ | 0.281 | 0.162 | 0.268 | 0.260 | 0.161 | 0.063 | 0.093 | 0.184 |
| $3.7 \times 10^6$ | 0.287 | 0.176 | 0.249 | 0.289 | 0.234 | 0.097 | 0.126 | 0.208 |
| $3.7 \times 10^6$ | 0.294 | 0.179 | 0.253 | 0.292 | 0.236 | 0.096 | 0.128 | 0.211 |
| $3.7 \times 10^6$ | 0.291 | 0.176 | 0.259 | 0.289 | 0.236 | 0.099 | 0.129 | 0.211 |
| $3.7 \times 10^6$ | 0.297 | 0.183 | 0.256 | 0.291 | 0.232 | 0.094 | 0.125 | 0.211 |
| $3.7 \times 10^6$ | 0.289 | 0.177 | 0.259 | 0.289 | 0.236 | 0.099 | 0.130 | 0.211 |
| $1.4 \times 10^6$ | 0.317 | 0.211 | 0.257 | 0.318 | 0.274 | 0.114 | 0.129 | 0.231 |
| $1.4 \times 10^6$ | 0.308 | 0.208 | 0.258 | 0.316 | 0.277 | 0.118 | 0.132 | 0.231 |
| $1.4 \times 10^6$ | 0.310 | 0.205 | 0.259 | 0.318 | 0.273 | 0.118 | 0.132 | 0.231 |
| $1.4 \times 10^6$ | 0.311 | 0.205 | 0.263 | 0.318 | 0.266 | 0.120 | 0.135 | 0.231 |
| $1.4 \times 10^6$ | 0.313 | 0.216 | 0.262 | 0.318 | 0.269 | 0.117 | 0.136 | 0.233 |
| $9.3 \times 10^5$ | 0.340 | 0.232 | 0.278 | 0.341 | 0.300 | 0.128 | 0.140 | 0.251 |
| $9.3 \times 10^5$ | 0.332 | 0.238 | 0.281 | 0.339 | 0.298 | 0.128 | 0.140 | 0.251 |
| $9.3 \times 10^5$ | 0.331 | 0.233 | 0.279 | 0.342 | 0.296 | 0.132 | 0.141 | 0.250 |
| $9.3 \times 10^5$ | 0.339 | 0.234 | 0.282 | 0.342 | 0.300 | 0.135 | 0.146 | 0.254 |
| $9.3 \times 10^5$ | 0.331 | 0.228 | 0.280 | 0.340 | 0.298 | 0.129 | 0.142 | 0.250 |
| $2.3 \times 10^5$ | 0.410 | 0.383 | 0.347 | 0.420 | 0.353 | 0.247 | 0.240 | 0.343 |
| $2.3 \times 10^5$ | 0.408 | 0.377 | 0.347 | 0.418 | 0.349 | 0.234 | 0.234 | 0.338 |
| $2.3 \times 10^5$ | 0.407 | 0.378 | 0.338 | 0.417 | 0.348 | 0.249 | 0.238 | 0.339 |
| $2.3 \times 10^5$ | 0.409 | 0.374 | 0.342 | 0.417 | 0.350 | 0.246 | 0.236 | 0.339 |
| $2.3 \times 10^5$ | 0.419 | 0.379 | 0.338 | 0.418 | 0.349 | 0.246 | 0.234 | 0.340 |
| $1.4 \times 10^5$ | 0.427 | 0.390 | 0.360 | 0.431 | 0.353 | 0.319 | 0.300 | 0.369 |
| $1.4 \times 10^5$ | 0.430 | 0.393 | 0.373 | 0.431 | 0.359 | 0.326 | 0.302 | 0.373 |
| $1.4 \times 10^5$ | 0.429 | 0.398 | 0.362 | 0.432 | 0.358 | 0.329 | 0.301 | 0.373 |
| $1.4 \times 10^5$ | 0.431 | 0.390 | 0.364 | 0.429 | 0.353 | 0.329 | 0.301 | 0.371 |
| $1.4 \times 10^5$ | 0.429 | 0.397 | 0.374 | 0.431 | 0.358 | 0.327 | 0.299 | 0.374 |
| $5.8 \times 10^4$ | 0.470 | 0.478 | 0.446 | 0.459 | 0.390 | 0.525 | 0.405 | 0.453 |
| $5.8 \times 10^4$ | 0.469 | 0.470 | 0.447 | 0.458 | 0.386 | 0.520 | 0.400 | 0.450 |
| $5.8 \times 10^4$ | 0.467 | 0.475 | 0.450 | 0.459 | 0.387 | 0.518 | 0.400 | 0.451 |
| $5.8 \times 10^4$ | 0.468 | 0.469 | 0.442 | 0.458 | 0.390 | 0.525 | 0.401 | 0.450 |
| $5.8 \times 10^4$ | 0.468 | 0.473 | 0.453 | 0.462 | 0.396 | 0.533 | 0.406 | 0.456 |
| $1.4 \times 10^4$ | 0.497 | 0.494 | 0.486 | 0.494 | 0.443 | 0.620 | 0.473 | 0.501 |
| $1.4 \times 10^4$ | 0.498 | 0.507 | 0.479 | 0.492 | 0.441 | 0.624 | 0.473 | 0.502 |
| $1.4 \times 10^4$ | 0.497 | 0.506 | 0.481 | 0.491 | 0.442 | 0.611 | 0.470 | 0.500 |
| $1.4 \times 10^4$ | 0.498 | 0.500 | 0.488 | 0.492 | 0.439 | 0.615 | 0.469 | 0.500 |
| $1.4 \times 10^4$ | 0.497 | 0.508 | 0.482 | 0.493 | 0.437 | 0.624 | 0.473 | 0.502 |

Table 19: Full experimental results measured by EER for varying *model sizes*. The average performance is shown in Fig. 5 (Right Bottom).

| EER | DFD | CDF V2 | Wild | Forgery. | DFF | DF40 | ScaleDF | Mean |
|---|---|---|---|---|---|---|---|---|
| 21.7M | 0.293 | 0.172 | 0.270 | 0.304 | 0.264 | 0.085 | 0.130 | 0.217 |
| 38.3M | 0.309 | 0.175 | 0.269 | 0.285 | 0.228 | 0.077 | 0.120 | 0.209 |
| 85.8M | 0.281 | 0.162 | 0.268 | 0.260 | 0.161 | 0.063 | 0.093 | 0.184 |
| 303.4M | 0.271 | 0.149 | 0.260 | 0.225 | 0.114 | 0.048 | 0.072 | 0.163 |
| 630.8M | 0.270 | 0.162 | 0.277 | 0.212 | 0.082 | 0.050 | 0.047 | 0.157 |

Table 20: Visualization of processed faces in ScaleDF.

| Name | Demo 0 | Demo 1 | Demo 2 | Demo 3 | Demo 4 |
|------|--------|--------|--------|--------|--------|
| GRID | | | | | |
| MORPH-2 | | | | | |
| LFW | | | | | |
| Multi-PIE | | | | | |
| GENKI-4K | | | | | |
| YouTubeFaces | | | | | |
| IMFDB | | | | | |
| Adience | | | | | |
| CACD | | | | | |
| CASIA-WebFace | | | | | |

Table 21: Visualization of processed faces in ScaleDF.

| Name | Demo 0 | Demo 1 | Demo 2 | Demo 3 | Demo 4 |
|------|--------|--------|--------|--------|--------|
| CREMA-D | | | | | |
| FaceScrub | | | | | |
| 300VW | | | | | |
| CelebA | | | | | |
| AFAD | | | | | |
| CFPW | | | | | |
| WIDER FACE | | | | | |
| AffectNet | | | | | |
| AgeDB | | | | | |
| MAFA | | | | | |

Table 22: Visualization of processed faces in ScaleDF.

| Name | Demo 0 | Demo 1 | Demo 2 | Demo 3 | Demo 4 |
|------|--------|--------|--------|--------|--------|
| RAF-DB | | | | | |
| UMDFaces | | | | | |
| UTKFace | | | | | |
| AFEW-VA | | | | | |
| MegaAge | | | | | |
| AVA | | | | | |
| AVSpeech | | | | | |
| ExpW | | | | | |
| IMDb-Face | | | | | |
| RAVDESS | | | | | |

Table 23: Visualization of processed faces in ScaleDF.

| Name | Demo 0 | Demo 1 | Demo 2 | Demo 3 | Demo 4 |
|------|--------|--------|--------|--------|--------|
| Tufts Face | | | | | |
| VGGFace2 | | | | | |
| Celeb-500K | | | | | |
| IJB-C | | | | | |
| VoxCeleb2 | | | | | |
| Aff-Wild2 | | | | | |
| FFHQ | | | | | |
| FaceForensics++ | | | | | |
| BUPT-CBFace | | | | | |
| DFEW | | | | | |

Table 24: Visualization of processed faces in ScaleDF.

| Name | Demo 0 | Demo 1 | Demo 2 | Demo 3 | Demo 4 |
|------|--------|--------|--------|--------|--------|
| MEAD | | | | | |
| MMA | | | | | |
| SAMM V3 | | | | | |
| FairFace | | | | | |
| Glint360K | | | | | |
| SpeakingFaces | | | | | |
| Wiki-Faces | | | | | |
| Asian-Celeb | | | | | |
| CelebV-HQ | | | | | |
| RMFD | | | | | |

Table 25: Visualization of processed faces in ScaleDF.

| Name | Demo 0 | Demo 1 | Demo 2 | Demo 3 | Demo 4 |
|------|--------|--------|--------|--------|--------|
| FaceVid-1K | | | | | |
| Faceswap | | | | | |
| FaceSwap | | | | | |
| DeepFakes | | | | | |
| FSGAN | | | | | |
| SimSwap | | | | | |
| HifiFace | | | | | |
| InfoSwap | | | | | |
| UniFace | | | | | |
| MobileFaceSwap | | | | | |

Table 26: Visualization of processed faces in ScaleDF.

| Name | Demo 0 | Demo 1 | Demo 2 | Demo 3 | Demo 4 |
|------|--------|--------|--------|--------|--------|
| E4S | | | | | |
| GHOST | | | | | |
| BlendFace | | | | | |
| FaceDancer | | | | | |
| 3DSwap | | | | | |
| Inswapper | | | | | |
| FaceAdapter | | | | | |
| CSCS | | | | | |
| REFace | | | | | |
| FaceFusion | | | | | |

Table 27: Visualization of processed faces in ScaleDF.

| Name | Demo 0 | Demo 1 | Demo 2 | Demo 3 | Demo 4 |
|------|--------|--------|--------|--------|--------|
| InstantID | | | | | |
| DiffFace | | | | | |
| Face2Face | | | | | |
| FOMM | | | | | |
| NeuralTextures | | | | | |
| OneShot | | | | | |
| Face-Vid2Vid | | | | | |
| TPSMM | | | | | |
| DaGAN | | | | | |
| LIA | | | | | |

Table 28: Visualization of processed faces in ScaleDF.

| Name | Demo 0 | Demo 1 | Demo 2 | Demo 3 | Demo 4 |
|------|--------|--------|--------|--------|--------|
| AMatrix | | | | | |
| StyleMask | | | | | |
| MRFA | | | | | |
| HyperReenact | | | | | |
| MCNet | | | | | |
| CVTHead | | | | | |
| FollowYourEmoji | | | | | |
| LivePortrait | | | | | |
| Megactor | | | | | |
| G3FA | | | | | |

Table 29: Visualization of processed faces in ScaleDF.

| Name | Demo 0 | Demo 1 | Demo 2 | Demo 3 | Demo 4 |
|---|---|---|---|---|---|
| FSRT | | | | | |
| SkyReels-A1 | | | | | |
| StyleGAN2 | | | | | |
| VQGAN | | | | | |
| StyleGAN3 | | | | | |
| StyleGAN-XL | | | | | |
| SD2.1 | | | | | |
| SD1.5 | | | | | |
| SDXL | | | | | |
| PixArt-Alpha | | | | | |

Table 30: Visualization of processed faces in ScaleDF.

| Name | Demo 0 | Demo 1 | Demo 2 | Demo 3 | Demo 4 |
|------|--------|--------|--------|--------|--------|
| Midjourney | | | | | |
| SD3.5 | | | | | |
| FLUX.1 [dev] | | | | | |
| CogView4 | | | | | |
| CogView3 | | | | | |
| Kolors | | | | | |
| Hunyuan-DiT | | | | | |
| LTX-Video | | | | | |
| HunyuanVideo | | | | | |
| Pika | | | | | |

Table 31: Visualization of processed faces in ScaleDF.

| Name | Demo 0 | Demo 1 | Demo 2 | Demo 3 | Demo 4 |
|------|--------|--------|--------|--------|--------|
| GPT-Image-1 | | | | | |
| Janus-Pro | | | | | |
| SimpleAR | | | | | |
| Wan-T2V | | | | | |
| Pyramid Flow | | | | | |
| CogVideoX | | | | | |
| SDEdit | | | | | |
| E4E | | | | | |
| EDICT | | | | | |
| DiffusionCLIP | | | | | |

Table 32: Visualization of processed faces in ScaleDF.

| Name | Demo 0 | Demo 1 | Demo 2 | Demo 3 | Demo 4 |
|------|--------|--------|--------|--------|--------|
| VecGAN | | | | | |
| InstructPix2Pix | | | | | |
| IP-Adapter | | | | | |
| MaskFaceGAN | | | | | |
| SDFlow | | | | | |
| EmoStyle | | | | | |
| Triplane | | | | | |
| FaceID | | | | | |
| AnySD | | | | | |
| MagicFace | | | | | |

Table 33: Visualization of processed faces in ScaleDF.

| Name | Demo 0 | Demo 1 | Demo 2 | Demo 3 | Demo 4 |
|------|--------|--------|--------|--------|--------|
| RigFace | | | | | |
| FluxEdit | | | | | |
| RFInversion | | | | | |
| Step1X-Edit | | | | | |
| MakeItTalk | | | | | |
| Wav2Lip | | | | | |
| Audio2Head | | | | | |
| SadTalker | | | | | |
| Video-Retalking | | | | | |
| DreamTalk | | | | | |

Table 34: Visualization of processed faces in ScaleDF.

| Name | Demo 0 | Demo 1 | Demo 2 | Demo 3 | Demo 4 |
|------|--------|--------|--------|--------|--------|
| IP_LAP | | | | | |
| Real3DPortrait | | | | | |
| FLOAT | | | | | |
| JoyVASA | | | | | |
| DAWN | | | | | |
| AniTalker | | | | | |
| AniPortrait | | | | | |
| EDTalk | | | | | |
| Diff2Lip | | | | | |
| JoyHallo | | | | | |

Table 35: Visualization of processed faces in ScaleDF.

| Name | Demo 0 | Demo 1 | Demo 2 | Demo 3 | Demo 4 |
|------|--------|--------|--------|--------|--------|
| Ditto | | | | | |
| KDTalker | | | | | |
| Echomimic | | | | | |

Table 36: Demonstration of the perturbations used when training models on the ScaleDF dataset.

| Perturb | Elaboration | Demo 0 | Demo 1 | Demo 2 | Demo 3 |
|---|---|---|---|---|---|
| ResizeCrop | Randomly crop and re-size an image to a specified size. | | | | |
| ColorJitter | Randomly change the brightness, contrast, satu-ration, and hue of an image. | | | | |
| Blur | Randomly ap-ply a blur filter to an image. | | | | |
| Pixelate | Pixelate ran-dom portions of an image. | | | | |
| Rotate | Randomly ro-tate an image within a given range of de-grees. | | | | |
| GrayScale | Convert an image into grayscale. | | | | |
| Padding | Pad an image with random colors, height and width. | | | | |
| AddNoise | Add random noise to an image. | | | | |
| VertFlip | Flip an image vertically. | | | | |
| HoriFlip | Flip an image horizontally. | | | | |

Table 37: Demonstration of the perturbations used when training models on the ScaleDF dataset.

| Perturb | Elaboration | Demo 0 | Demo 1 | Demo 2 | Demo 3 |
|---------|-------------|--------|--------|--------|--------|
| PerspChange | Randomly transform the perspective of an image. | | | | |
| ChangeChan | Randomly shift, swap, or invert the channel of an image. | | | | |
| EncQuality | Randomly encode (reduce) the quality of an image. | | | | |
| Sharpen | Randomly enhance the edge contrast of an image. | | | | |
| Skew | Randomly skew an image by a certain angle. | | | | |
| ShufPixels | Randomly rearrange (shuffle) the pixels within an image. | | | | |
| Xraylize | Simulate the effect of an X-ray on an image. | | | | |
| GlassEffect | Add glass effect onto an image with random extent. | | | | |
| OptDistort | View an image through a medium that randomly distorts the light. | | | | |
| Solarize | Invert all pixel values above a random threshold. | | | | |

Table 38: Demonstration of the perturbations used when training models on the ScaleDF dataset.

| Perturb | Elaboration | Demo 0 | Demo 1 | Demo 2 | Demo 3 |
|---------|-------------|--------|--------|--------|--------|
| Zooming | Randomly simulate the effect of zooming in or out. | | | | |
| Elastic | Simulate a jelly-like distortion of an image. | | | | |
| FancyPCA | Random use PCA to alter the intensities of the RGB channels. | | | | |
| GridDistort | Apply random non-linear distortions within a grid to an image. | | | | |
| ISONoise | Apply random camera sensor noise to an image. | | | | |
| Multiple | A random value, is multiplied with the pixel values of an image. | | | | |
| Posterize | Randomly reduce the number of bits of each pixel. | | | | |
| Gamma | Alter the luminance values of an image by applying a power-law function. | | | | |
| Spatter | Randomly create a spatter effect on an image. | | | | |
| Binary | Use different methods to binarize an image. | | | | |

