# OpenReview forum: "Scaling Laws for Deepfake Detection"
_ICLR.cc/2026/Conference — ICLR 2026 Conference Withdrawn Submission_

### Official Review · Reviewer_McQc · 2025-10-26

**Soundness:** 3
**Presentation:** 3
**Contribution:** 2
**Rating:** 4
**Confidence:** 4

**Summary:**

In this paper, the authors conduct an empirical study of scaling laws for deepfake detection. To this end, the authors first construct ScaleDF, a dataset comprising over 5.8 million real face images from 51 domains and over 8.8 million fake images generated by 102 deepfake methods. Then, the authors investigate how model performance scales with the number of real domains, deepfake methods, and training images, observing power-law relationships akin to those in LLMs.

Overall, this paper offers meaningful insights into the development of future deepfake detectors. Nevertheless, the analysis of scaling law phenomena is somewhat constrained, more comprehensive and diverse experimental investigations are needed.

**Strengths:**

1. The authors propose ScaleDF, the biggest deepfake dataset which contains 51 real domains and 102 deepfake methods, the dataset far exceeds the scale and diversity of prior deepfake detection datasets, addressing a critical bottleneck for robust experimental analysis.

2. The authors conduct a thorough study of scaling laws in deepfake detection. It empirically validates power-law scaling for AUC with respect to the number of real domains and deepfake methods.

3. The authors use clear mathematical formulations (e.g., $1-\bar{AUC}=A \cdot N^{-\alpha}$ for scaling laws), and provide extensive ablations and benchmarking to support their claims.

**Weaknesses:**

1.  The core contributions of this paper are largely around dataset compilation and large-scale experimentation rather than providing new algorithms, theoretical insights, or methodological innovations for deepfake detection.

2. The experiments are mainly based on standard ViT backbones, without exploring whether alternate architectures, more nuanced objective functions, or specialized training regimes (e.g., domain adaptation or contrastive representation learning) might better leverage data diversity.

3. The results generally present average AUC or EER. While this is appropriate for summarizing trends, it lacks fine-grained details, such as how scaling affects different categories of deepfake (e.g., face swapping vs. attribute editing).

4. Domain leakage should be considered. Although some effort is made to avoid overfitting, at such a scale, there is inevitable dataset overlap across splits. The measures taken for “cross-domain” or “cross-method” testing are described, yet quantification of leakage is missing.

5. The scaling law is observed with empirical results. There is little attempt to model, explain, or theoretically motivate why these (double-)power laws arise in deepfake detection, i.e., no mechanistic or statistical argument is proposed to connect data diversity, method coverage, or architecture capacity. For instance, the choice of power law forms such as $1-\bar{AUC}=A \cdot N^{-\alpha}$ is motivated mainly by empirical fit. Moreover, some thresholds are left as extrapolations (e.g., “300 domains to reach 0.95 AUC”), without uncertainty quantification.

**Questions:**

Please see Weaknesses

---

### Official Review · Reviewer_47SP · 2025-10-26

**Soundness:** 2
**Presentation:** 3
**Contribution:** 2
**Rating:** 2
**Confidence:** 4

**Summary:**

This paper presents an empirical study of scaling laws for the deepfake detection task. Specifically, the authors collect a large dataset consisting of over 5.8 real images and 8.8 million fake images from deepfake models, and formalize the task as a binary classification problem. The authors further test the performance using different deepfake detection methods on this proposed ScaleDF dataset, and propose several empirical observations of the scaling behaviors from those models, such as the power-law scaling w.r.t. the number of training real domains and deepfake methods.

**Strengths:**

- The problem of deepfake detection is an important research topic given the current popularity and advances of deep generative models.

- The paper has a clear structure and makes it easy to follow.

**Weaknesses:**

- From a problem formulation perspective, while deepfake detection is a valuable research direction, the paper approaches it as a binary classification task in a somewhat oversimplified manner. In practice, current generative methods often perform fine-grained editing on only small regions of an image, which makes detection more nuanced. As a result, the paper’s formulation may limit its applicability to more realistic and challenging scenarios, narrowing its potential downstream impact, as an empirical analytical work.

- From the experimental perspective, the paper lacks comparison on several recent important detection methods proposed under the similar binary classification and scaled-up setting, such as [a].

- The paper could benefit from a more in-depth analysis. As it stands, the evaluation lacks detailed breakdowns and mostly presents high-level observations—such as “scaling helps”, which, while valid, are relatively intuitive and offer limited new insight. A more granular analysis, such as domain-wise or manipulation-type-specific performance, could yield more meaningful takeaways.

- Additionally, as the authors acknowledge, the dataset may raise ethical and copyright concerns due to its inclusion of a large number of deepfake face images. I would recommend that ethical reviewers examine this aspect closely.

- I would advise more caution regarding the “in-the-wild” setting described in Section 3.1. Benchmarks like Celeb-DF V2 and ForgeryNet may contain images that overlap with the training data in the proposed dataset, potentially inflating evaluation results. Further validation is needed to support the robustness of the in-the-wild claims.

---
[a] D^3: Scaling Up Deepfake Detection by Learning from Discrepancy. In CVPR 2025.

**Questions:**

The paper presents several weaknesses, in my opinion, that are challenging to clarify and resolve under the current framework. I therefore do recommend the paper acceptance at this stage. Please refer to the Weaknesses for details.

**Details Of Ethics Concerns:**

The dataset may raise ethical and copyright concerns due to its inclusion of a large number of deepfake face images. I would recommend that ethical reviewers examine this aspect closely.

---

### Official Review · Reviewer_c9vH · 2025-10-29

**Soundness:** 3
**Presentation:** 3
**Contribution:** 3
**Rating:** 4
**Confidence:** 4

**Summary:**

This paper introduces ScaleDF, a large dataset for deepfake detection that includes 51 real domains, 102 forgery methods, and 14 million images. Using this dataset, the authors study how detection performance scales with the number of real domains, fake methods, and images, finding clear power-law trends. The study is careful and the dataset is useful, but it mostly confirms scaling behaviors already known from image classification, without adding new theory or model ideas.

**Strengths:**

1. The authors conduct a careful and systematic investigation of scaling laws for deepfake detection, varying three dimensions: the number of real domains, forgery methods, and image samples. Each scaling relationship is fit using well-defined power-law or double-saturating models, with high coefficients of determination (R² > 0.98). This rigorous quantitative approach is consistent with prior scaling studies such as Kaplan et al. (2020, OpenAI), showing a commendable level of experimental thoroughness and statistical discipline.

2. The proposed ScaleDF dataset (51 real domains, 102 forgery methods, and 14 million images) is an impressive engineering contribution that significantly exceeds prior benchmarks like ForgeryNet and DF40. This large-scale dataset substantially improves coverage across different generation paradigms (GANs, Diffusion, Flow Matching, etc.) and offers a strong foundation for future research on data-driven forensic generalization. Its scope alone makes it a valuable community resource even if the methodological contributions are limited.

3. The paper is well-written, with transparent documentation of the training settings, augmentation schemes, and evaluation protocols. The use of standard architectures and public baselines enhances reproducibility, and the authors provide enough implementation details to replicate their findings. Such openness aligns with reproducibility practices.

**Weaknesses:**

1. The study provides solid empirical results, but the theoretical novelty is quite limited. The observed scaling laws largely mirror patterns already well-documented in earlier literature. The functional forms, such as (1 - \text{AUC} = A N^{-\alpha}) and the double-saturating power law, directly follow existing analyses. The paper does not attempt to offer a new theoretical explanation for the scaling exponents or their transitions, so the contribution feels more like a domain-specific replication than a conceptual advance. While the authors present neat empirical curves, they stop short of explaining why these scaling behaviors emerge.

2. Much of the contribution overlaps with what is already known from image classification scaling. Since deepfake detection here is essentially a binary image classification task trained on standard ViT architectures, it is unsurprising that the same scaling behaviors appear. This replication of known trends, similar to  prior conclusion reported for vision transformers, makes the findings somewhat expected. The paper would have been stronger if it explored conditions under which deepfake detection behaves differently from general image classification, or if it revealed forensic-specific scaling effects.

3. The scope is confined to static image detection and overlooks temporal or multimodal dynamics. All experiments focus on single-frame image analysis, leaving out the temporal and cross-modal cues that are central to practical deepfake forensics. It would be possible that multimodal scaling behaves differently and reveal richer dynamics. By excluding these aspects, the paper captures only part of the scaling landscape, reducing the generality of its conclusions for real-world multimodal scenarios.

**Questions:**

1.The authors may clarify whether any theoretical interpretation or modeling effort was made to explain the observed scaling exponents, beyond empirical curve fitting.
2.The paper should better distinguish deepfake detection from standard image classification in terms of scaling behavior and identify any conditions where the two might diverge.
3.The authors are encouraged to extend the scaling analysis to multimodal or temporal settings and discuss whether similar power-law behavior is expected in those scenarios.

---

### Official Review · Reviewer_o4oD · 2025-11-10

**Soundness:** 3
**Presentation:** 4
**Contribution:** 3
**Rating:** 4
**Confidence:** 4

**Summary:**

This paper proposes to study the scaling laws for deepfake detection, where a large scale dataset (named as ScaleDF) composed of 5.8M real face images from 51 different domains/datasets and 8.8M fake face images from 102 deepfake generation methods is built for supporting such study, in which the various perspectives of diversity are specifically considered. The study aims at the relation among deepfake detection model performance (in terms of AUC and EER) and the number of real image domains, deepfake generation methods, and training images, in which two main observations are drawn: 1) power-law scaling is found with respect to the number of training real domains and deepfake methods, but not the caused by a change in the number of real or fake images; 2) double-saturating power-law scaling is observed with respect to the number of training images.

**Strengths:**

+ The proposed dataset for studying the scaling laws for deepfake detection is extensive, large-scale, and built with systematical structure (e.g. containing various categories of deepfake generation methods, in which these categories are balanced).
+ The observations of scaling laws drawn from the experiments provide valuable rule-of-thumb for the following researches upon deepfake detection, e.g. focusing more on the diversity of generation method categories, as well as the importance of data augmentation in the context of scaling.

**Weaknesses:**

- The current target of deepfake detection only focuses on human faces, while there lacks for the discussion upon general images, and it is hard to estimate if the drawn observations can be applied to general images.
- As the current target of study is facial images, there exists data preprocessing pipeline particularly designed for faces (i.e. face detection, alignment, and cropping). Nevertheless, it is uncertain if the preprocessing operation (especially the alignment operation which has the transformation upon facial regions) would affect the investigation upon deepfake detection (the related question: why it is necessary to have the face frontalization to perform the deepfake detection).
- The current deepfake detection model used in this study is based on a ViT-based binary classifier built by the authors, it would be better to investigate if the same observations can be drawn while using any other existing deepfake detectors.
- In addition to discover the scaling laws, there should be further analysis upon the dependencies among the data distributions from different domains (e.g. for detector which has been trained upon face reenactment images, the improvement brought by samples from face attribute editing would be less) thus indicating the potential shortfall/gap in the entire training data distribution which should be filled up in the coming future (i.e. the region in the data distribution which would be more beneficial/informative for learning the detector).

**Questions:**

The paper is well organized and presented, together with the proposed extensive dataset to study the scaling laws for deepfake detection problem as well as the corresponding valuable observations drawn from the experimental results. Nevertheless, there are some concerns which should be addressed in the rebuttal to further enhance the quality of the submission, including the discussion upon deepfake detection for general images, the influence made by processing operations, the experiments based on more existing detection methods, and the analysis among the generated data distribution from various method categories.

---

### Note · Authors · 2025-11-18

I have read and agree with the venue's withdrawal policy on behalf of myself and my co-authors.